# Hydrogenation versus hydrogenolysis during alkaline electrochemical valorization of 5-hydroxymethylfurfural over oxide-derived Cu-bimetallics

Philipp Hauke[1], Thomas Merzdorf[1], Malte Klingenhof[1] & Peter Strasser [1]✉

The electrochemical conversion of 5-Hydroxymethylfurfural, especially its reduction, is an attractive green production pathway for carbonaceous e-chemicals. We demonstrate the reduction of 5-Hydroxymethylfurfural to 5-Methylfurfurylalcohol under strongly alkaline reaction environments over oxide-derived Cu bimetallic electrocatalysts. We investigate whether and how the surface catalysis of the $MO_x$ phases tune the catalytic selectivity of oxide-derived Cu with respect to the 2-electron hydrogenation to 2.5-Bishydroxymethylfuran and the (2 + 2)-electron hydrogenation/hydrogenolysis to 5-Methylfurfurylalcohol. We provide evidence for a kinetic competition between the evolution of $H_2$ and the 2-electron hydrogenolysis of 2.5-Bishydroxymethylfuran to 5-Methylfurfurylalcohol and discuss its mechanistic implications. Finally, we demonstrate that the ability to conduct 5-Hydroxymethylfurfural reduction to 5-Methylfurfurylalcohol in alkaline conditions over oxide-derived Cu/$MO_x$ Cu foam electrodes enable an efficiently operating alkaline exchange membranes electrolyzer, in which the cathodic 5-Hydroxymethylfurfural valorization is coupled to either alkaline oxygen evolution anode or to oxidative 5-Hydroxymethylfurfural valorization.

The electrochemical conversion of biomass feedstock under near-ambient operation conditions inside membrane electrolyzers using renewable energy is emerging as an attractive alternative production pathway for carbonaceous green e-chemicals. In particular, the biomass compound 5-Hydroxymethylfurfural (HMF)—with its functional aldehyde and alcohol groups at positions 2 and 5 of the furan ring (Fig. 1a)—is a promising platform molecule for an electrochemical oxidative and reductive valorization to bio-based polymer building blocks and fuel additives, respectively. While the electrochemical oxidation of HMF is well documented[1–4], the HMF reduction reaction (HMFRR) received far less attention over the past years. In fact, the low kinetic barriers and low electrochemical overpotentials of the hydrogen evolution reaction (HER) make the competitive HMFRR process in

an aqueous environment rather unattractive. However, electrochemical reductive ring-opening, hydrogenation, and hydrogenolysis of HMF represent attractive future electricity-based reaction process pathways to agrochemicals, pharmaceuticals, bio-fuels, or polyesters[5]. To date, the electrochemical HMFRR has been largely reported over metallic Cu-based catalysts due to their unfavorable competitive HER performance, coupled to their high chemical affinity to organic molecules. In addition to the selective hydrogenation of HMF using noble metal catalysts, Roylance et al. showed the reductive ring opening over a Zn catalyst, while Kloth et al. showed the dimerization of HMF over carbon electrodes[6–9]. Bimetallic HMF reduction electrocatalysts, in particular bimetallic oxides in the form of their oxide-derived surface-roughened catalyst analogs that evolve under

[1]The Electrochemical Energy, Catalysis, and Materials Science Laboratory, Department of Chemistry, Chemical Engineering Division, Technical University Berlin, Berlin, Germany. ✉e-mail: pstrasser@tu-berlin.de

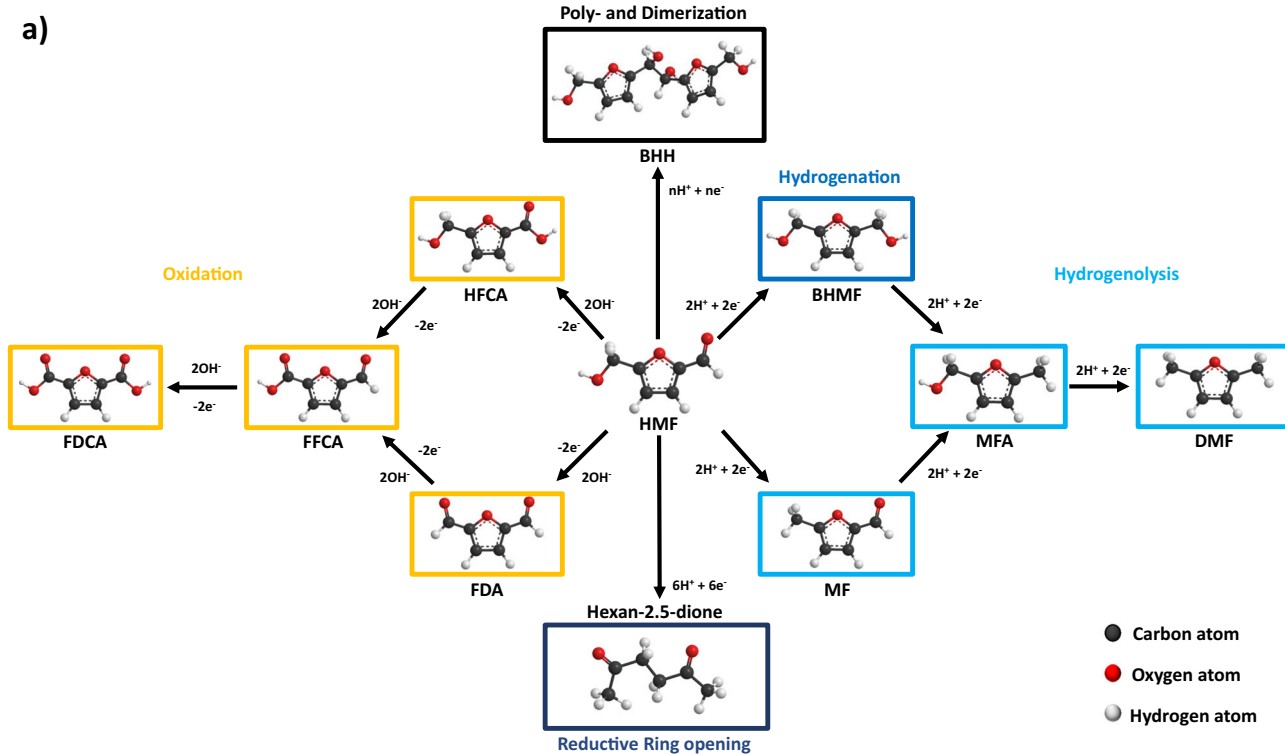

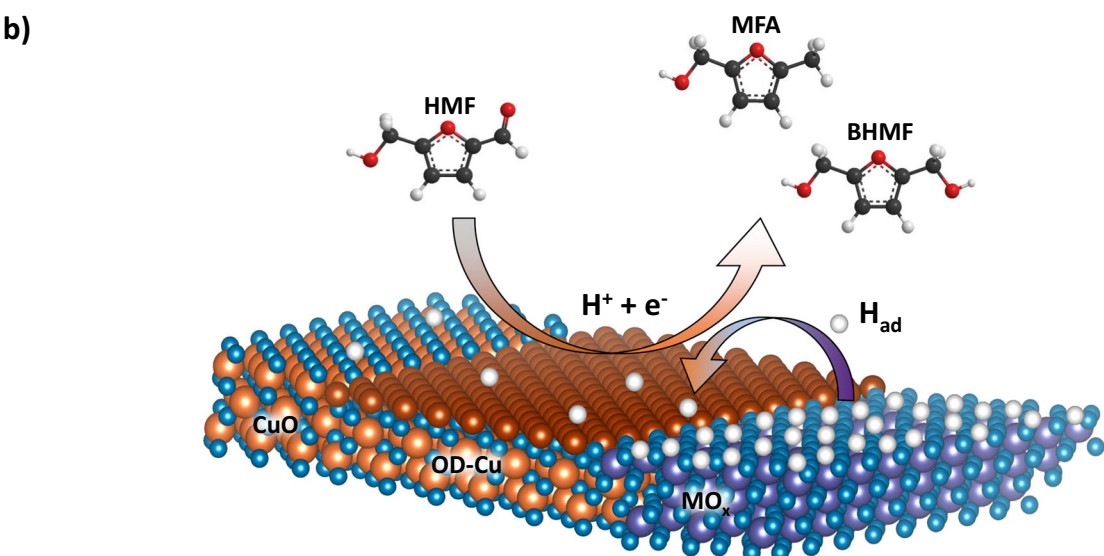

**Fig. 1 | HMF reaction pathways and two-phase Cu-based bimetallic oxide catalyst concept for enhance HMFRR. a** Poly- and Dimerization (black): undesired side reaction of n HMF molecules and n Protons/electrons. Reductive Ring opening (dark blue): opening of the furan ring with 6e⁻ and 6H⁺. Hydrogenation (blue): conversion of HMF to BHMF with 2e⁻ and 2H⁺. Hydrogenolysis (blue): conversion of HMF to MF with 2e⁻ and 2H⁺, conversion of BHMF or MF to MFA with 2e⁻ and 2H⁺, and conversion of MFA to DMF with 2e⁻ and 2H⁺. Oxidation (yellow): conversion of HMF over HFCA or FDA to FFCA and FDCA (6e⁻ and 6OH⁻). Carbon atoms (black), oxygen atoms (red), and hydrogen atoms (white). **b** Co-precipitated co-existing Tenorite CuO/MOₓ (Hematite Fe₂O₃) nanoparticles at the nm-scale are hypothesized to offer enhanced catalytic HMFRR reactivity after in-situ reduction to OD-Cu/MOₓ mixed phase catalysts. Adsorbed hydrogen atoms (H_ad) are given in white.

reducing *operando* conditions, are essentially unexplored. Prior work on the HMF reaction pathways seemingly established that the electroreduction of HMF beyond the first 2e⁻ hydrogenation product, 2.5-Bishydroxymethylfuran (BHMF), is not possible at high pH conditions. The rate of subsequent hydrogenolysis of BHMF to 5-Methylfurfurylalcohol (MFA) or 2.5-Dimethylfuran (DMF) sharply decreased with increasing pH[10,11]. Only in strongly acidic electrolytes,

work by Nilges et al. and Zhang et al. yielded the formation of MFA and even DMF via 4 and 6e⁻ HMF hydrogenolysis[12,13].

In this work, we show that, in contrast to the long-held view in literature, the valorization of HMF to MFA at high pH via the 4-electron coupled hydrogenation/hydrogenolysis is actually possible. We demonstrate that these conditions enable the use of CuO/MOₓ (M = first row transition metals) bimetallic oxide electrocatalyst precursors,

which—under the reductive reaction conditions—transform into operating oxide-derived partially metallic (OD)-Cu/$MO_x$ catalysts. In particular, the metal oxides $NiO$, $Fe_2O_3$, and $Co_3O_4$ were added to $CuO$ to tune the resulting HER activity of the two-phase system and by means of tuning the surface atomic $H_{ad}$ coverage. It is, to the best of our knowledge, the first report of electrocatalyst and electrochemical interfaces that are able to catalyze the HMF reduction to MFA in alkaline conditions. Mechanistically, we clarify whether and how the surface catalysis of the $MO_x$ phases tune the catalytic selectivity of OD-Cu with respect to the 2-electron hydrogenation to BHMF and the (2 + 2)-electron hydrogenation/hydrogenolysis to MFA. Finally, we design a foam-supported CuO/$MO_x$ cathode and operate it inside an alkaline exchange membranes (AEM) HMF electrolyzer. Continuous reductive HMF valorization coupled with water oxidation and even oxidative HMF valorization is demonstrated.

## Results

### Catalyst synthesis

Four crystalline mono-metallic powder oxides and three crystalline Cu-bimetallic oxide powder electrocatalysts were prepared using a (co-)precipitation-calcination (air) protocol. The calcined co-precipitated materials were deliberately designed as two-phase oxides, consisting of the dominant Tenorite Copper(II) oxide phase mixed with the second $MO_x$, M = Ni, Fe, Co, oxide phase, henceforth referred to as CuO/$MO_x$ catalyst (see structural cartoon Fig. 1b). The choice of a two-phase catalyst concept originated from the basic good HMFRR reactivity of pure crystalline CuO that we intended to tune by the presence of a distinct second oxide phase at nm scale proximity (rather than by forming a new mixed oxide phase) with varying structure and chemisorption characteristics[14–17]. Each two-phase catalyst synthesis was individually developed and optimized such as to set the molar ratio of metal M to 10%, see Supplementary Table 1. To achieve this, inductively coupled plasma optical emission spectroscopy (ICP-OES) was used. The 10% molar ratio originated from a representative CuO/NiO structure-composition-performance study (Supplementary Fig. 1 and Supplementary Discussion I) where the 10 mol% $MO_x$ catalysts displayed the widest HMF-selective reduction activity window with a balanced HER reactivity.

### Characterizations

X-ray powder diffraction (XRD) analysis of pure oxides revealed the characteristic Bragg reflections of Tenorite CuO (C2/c; JCPDS: 01-089-2529, Fig. 2a) at 35.54°, 38.66°, 61.58°, 66.30° and 68.87°, Bunsenite NiO (Fm$\bar{3}$m; JCPDS: 00-002-1216, Fig. 2b) at 37.29°, 43.26°, 62.74°, 75.38°, and 79.86°, Hematite $Fe_2O_3$ (R$\bar{3}$c; JCPDS: 01-089-0599, Fig. 2c) at 24.16°, 33.20°, 35.65°, 49.50°, 54.12°, 62.49° and 64.05, and for the $Co_3O_4$ Spinel (Fd$\bar{3}$m; JCPDS: 00-043-1003, Fig. 2d) at 19.00°, 31.26°, 36.83°, 38.54°, 44.81°, 55.65°, 59.36° and 65.22. Figure 2e shows the gradual change in the XRD pattern of two-phase CuO/NiO for increasing NiO content from 0 mol% (bottom) to 100 mol% NiO (top). Data indicate a diffractive detection limit of NiO above 67 mol% that agrees with Fig. 2f, where exclusive CuO Bragg reflections were detectable at the 10 mol% Ni level.

To further investigate the surface chemical composition and the valence states, X-ray photoemission spectroscopy (XPS) of pure CuO and the bimetallic metal oxide powders was carried out. It provided further evidence of the presence of Cu(II)O, Ni(II)O, Fe(III)$_2$O$_3$, and Co(II/III)$_3$O$_4$ (Fig. 2g–j and Supplementary Figs. 2–5). In more detail, Fig. 2g and Supplementary Fig. 2 reveal characteristic peaks for oxidic Cu at 953.8 eV (2$p_{1/2}$) and 933.8 eV (2$p_{3/2}$) as well as strong $Cu^{2+}$ satellites (sat.) peaks at 962.3 and 941.6 eV[18,19]. In addition, the characteristic peaks for NiO (872.6 and 855.3 eV), oxidic Fe (723.9 and 711.4 eV), oxidic Co (795.2 and 779.7 eV) as well as $Ni^{2+}$ (853.7 and 872.6 eV), $Fe^{3+}$ (732.1 and 718.7 eV) and $Co^{2+/3+}$ (795.6, 780, and 782.2 eV) are observed (Fig. 2h–j and Supplementary Figs. 3–5)[20–24]. Besides, for the Ni 2$p_{3/2}$

signal (855.3 eV), γ-NiOOH was detected proportionally, which probably arose during the strongly alkaline synthesis and was not further oxidized to NiO (Supplementary Fig. 3c)[23,25,26].

Brunauer-Emmett-Teller (BET) analysis was used to estimate the surface area of the mono- and bimetallic oxide catalyst powders (Fig. 2k and Supplementary Table 2). CuO showed a relatively low surface area yet twice the surface area of commercial Tenorite, CuO (Sigma-Aldrich). The highest surface area was exposed by pure NiO followed by $Co_3O_4$ and $Fe_2O_3$. However, this trend was not valid for the bimetallic powders, where CuO/NiO showed lower surface areas than CuO/$Fe_2O_3$ and CuO/$Co_3O_4$.

### Morphology and composition transformations under reaction

To inspect the CuO/$MO_x$ catalyst morphologies, in pure powder form transmission electron microscopy (TEM and HR-STEM) was used. Scanning electron microscopy-energy dispersive X-ray analysis (SEM-EDX) was used to explore the morphology and composition before and after electrolytic reaction (ae) of the corresponding catalyst thin films spray-coated onto the CF. TEM and HR-STEM images did not reveal any distinct differences between the catalyst morphologies or other visible differences between the various CuO/$MO_x$ catalysts (Supplementary Figs. 6 and 7). Individual oxide particle sizes ranged in the 5 to 100 nm range, with particles forming linearly attached chains.

The SEM-EDX elemental analysis of the spray-coated CF-supported electrocatalyst films showed the expected contributions of elements like O, Cu, Ni, Fe, and Co (Supplementary Figs. 8–12). Contributions of C and F stemmed from the Nafion binder. Just like the TEM analysis, the SEM images (Supplementary Figs. 9–12a–d) confirmed the agglomeration of induvial oval-shaped metal oxide nanoparticles. Although no significant morphological changes were observed after electrolysis (Supplementary Figs. 9–12a–d), a dramatic compositional change was evident in all catalysts, more specifically a decreased molar O/Cu ratio (Supplementary Figs. 9–12g, h), indicating a partial reduction of the CuO catalyst phase to metallic Cu. This justifies perceiving the CuO/$MO_x$ as two-phase precursor catalysts that reductively transformed into a catalytically active oxide-derived OD-Cu/$MO_x$ catalysts. There was no evidence that the $MO_x$ phases were reduced to a metallic state during catalytic reaction. OD-Cu catalysts are well documented in the field of $CO_2$ reduction[16,27]. They offer a very rough Cu surface characterized by many undercoordinated Cu adatoms which likely serve as active sites for activation steps of reactant molecules[16,27,28]. For the CuO/NiO/CF catalyst, needles were visible after electrolysis (Supplementary Fig. 10c, d). KOH crystal formation was excluded by reference measurements in rinsed and ultrasonicated electrodes. Similarly, judged by the SEM-EDX profiles, we did not see any Potassium salt precipitation for any of the other catalysts (Supplementary Figs. 10h, 11h and 12h).

To convince ourselves of the formation of metallic OD-Cu phases under reactive electrode potentials, thin film XRD studies were carried out on the as-prepared CuO precursor catalyst (center and bottom Graph) and the resulting OD-Cu catalyst after catalytic HMFRR (top profile) in Supplementary Fig. 13a, b. Evidently, the (−111), (111) and (200) reflections of Tenorite CuO (orange) disappeared partially in favor of the emergence of (111) and (200) reflections of metallic OD-Cu (red) confirming our SEM-EDX results.

### Catalytic testing of rotating disk electrodes

In order to investigate the catalytic HMF reduction reactivity (not HMFRR selectivity) of the precursor CuO/$MO_x$ catalysts, we carried out electrochemical three-electrode RDE measurements using both the pure individual metal oxide phases (Fig. 3a, b, d, e) and the two-phase CuO/$MO_x$ catalysts (Fig. 3c, f) in presence (solid lines) and in absence (dashed lines) of 10 mM HMF reactant. Figure 3a–c reports geometric area-corrected current densities, whereas Fig. 3d–f shows BET surface-corrected current densities, a popular intrinsic measure of performance.

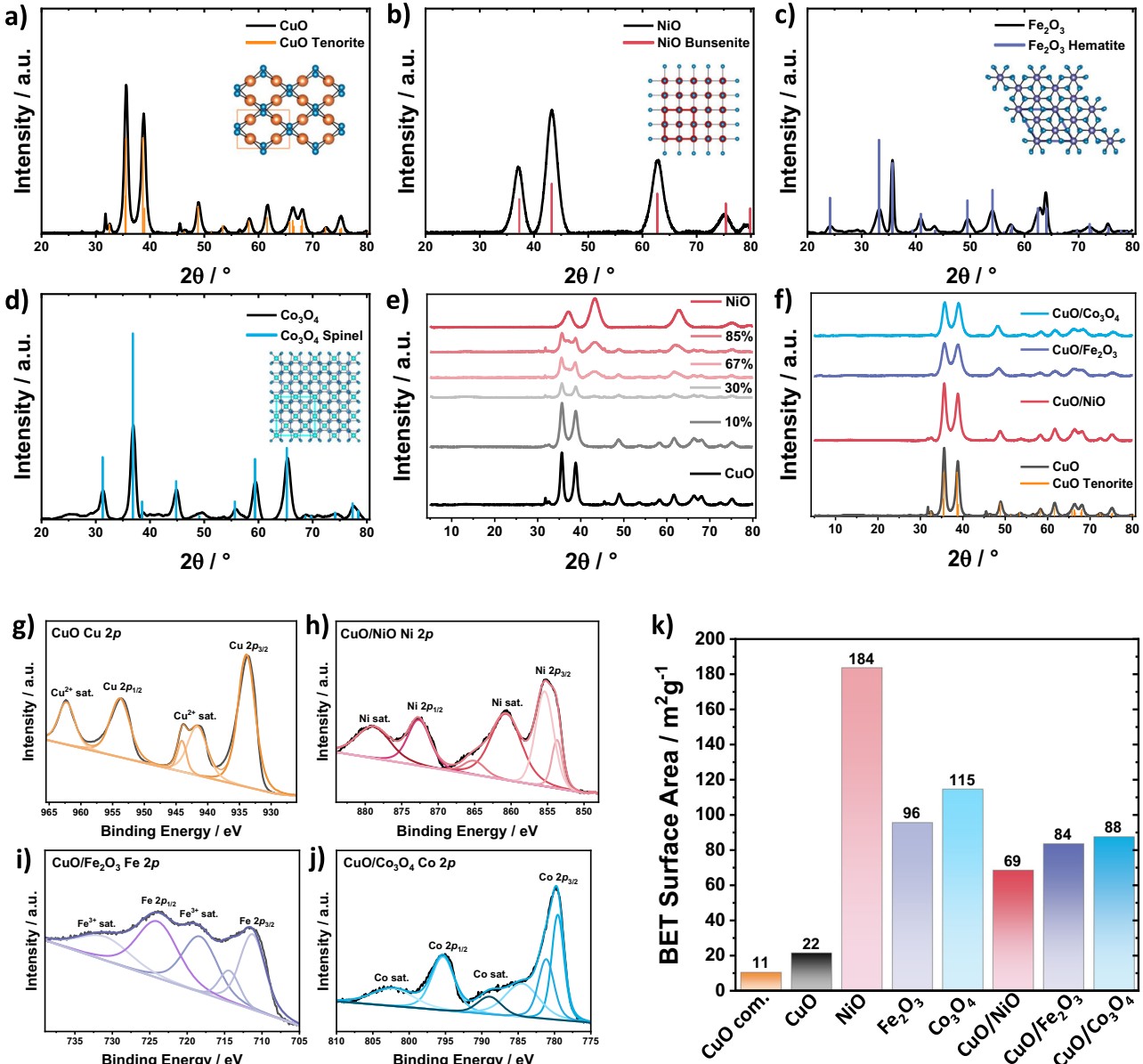

**Fig. 2 | XRD, XPS, and BET characterization of the CuO/MO$_x$ powder catalysts.** **a**–**d** XRD patterns of pure powder CuO (Tenorite), NiO (Bunsenite), Fe$_2$O$_3$ (Hematite), and Co$_3$O$_4$ (Spinel) XRD with references given in orange, red, purple, blue, and inserted crystal structures. **e** XRD patterns from pure powder CuO (black) over 10, 30, 67, and 80 mol% NiO to pure NiO (red). **f** XRD patterns of pure CuO (black), CuO/NiO (red), CuO/Fe$_2$O$_3$ (purple), and CuO/Co$_3$O$_4$ (blue). All CuO/MO$_x$ are presented with 10 mol% of the second metal oxide. CuO Tenorite reference is given in orange. **g** Cu 2p XPS measurements of pure CuO (orange). **h** Ni 2p XPS measurements of CuO/NiO (red). **i** Fe 2p XPS measurements of CuO/Fe$_2$O$_3$ (purple). **j** Co 2p XPS measurements of CuO/Co$_3$O$_4$ (blue). **k** BET results of commercial CuO (orange), synthesized CuO (black), pure NiO, Fe$_2$O$_3$, Co$_3$O$_4$ (light red, light purple, light blue), and mixed metal oxides CuO/NiO (red), CuO/Fe$_2$O$_3$ (purple), CuO/Co$_3$O$_4$ (blue).

We first addressed the pure oxide catalysts, CuO, NiO, Fe$_2$O$_3$, and Co$_3$O$_4$ (Fig. 3a), to learn more about their individual HMF reduction reactivity. Except for Fe$_2$O$_3$, the pure phase oxide catalysts displayed significantly enhanced catalytic HMF reduction (HMFRR, solid) over the hydrogen evolution reaction (HER, dashed). This was coupled to more anodic electrode potentials at 10 mA cm$^{-2}$ or likewise lower $\eta_{HMFRR}$ than $\eta_{HER}$. The operating OD-Cu catalyst (henceforth referred to by its precursor state CuO) by far outperformed other oxides (Fig. 3a), which was even more obvious in the BET-normalized plots (Fig. 3d). Despite evidence for the formation of OD-Cu and the existence of characteristic metal/metal oxides couple of NiO, Fe$_2$O$_3$, and Co$_3$O$_4$ (Supplementary Figs. 9–13), no voltammetric redox waves were obvious in base electrolyte without HMF[29–32]. Notably, the present crystalline oxide precursor catalysts showed substantially higher

catalytic activities than previously reported metallic catalysts[33,34], which we in part attribute to the higher surface roughness of the partially reduced oxide catalysts.

We also compared our synthesized crystalline Tenorite CuO catalysts to a commercial CuO material (Fig. 3b, e). The synthesized CuO catalyst showed higher geometric catalytic HMFRR, but similar intrinsic HMFRR as the commercial CuO (Fig. 3b, e); its HER performance was lower than the commercial one. This can be rationalized by distinct oxidic surface states, as oxide surfaces generally display weaker H chemisorption and hence lower H coverages. As the oxidic catalyst surface reduced to roughened metallic OD-Cu facets, the rough undercoordinated Cu$^0$ surface supported larger geometric HMF currents, yet the intrinsic activity per site remained comparable to the reference.

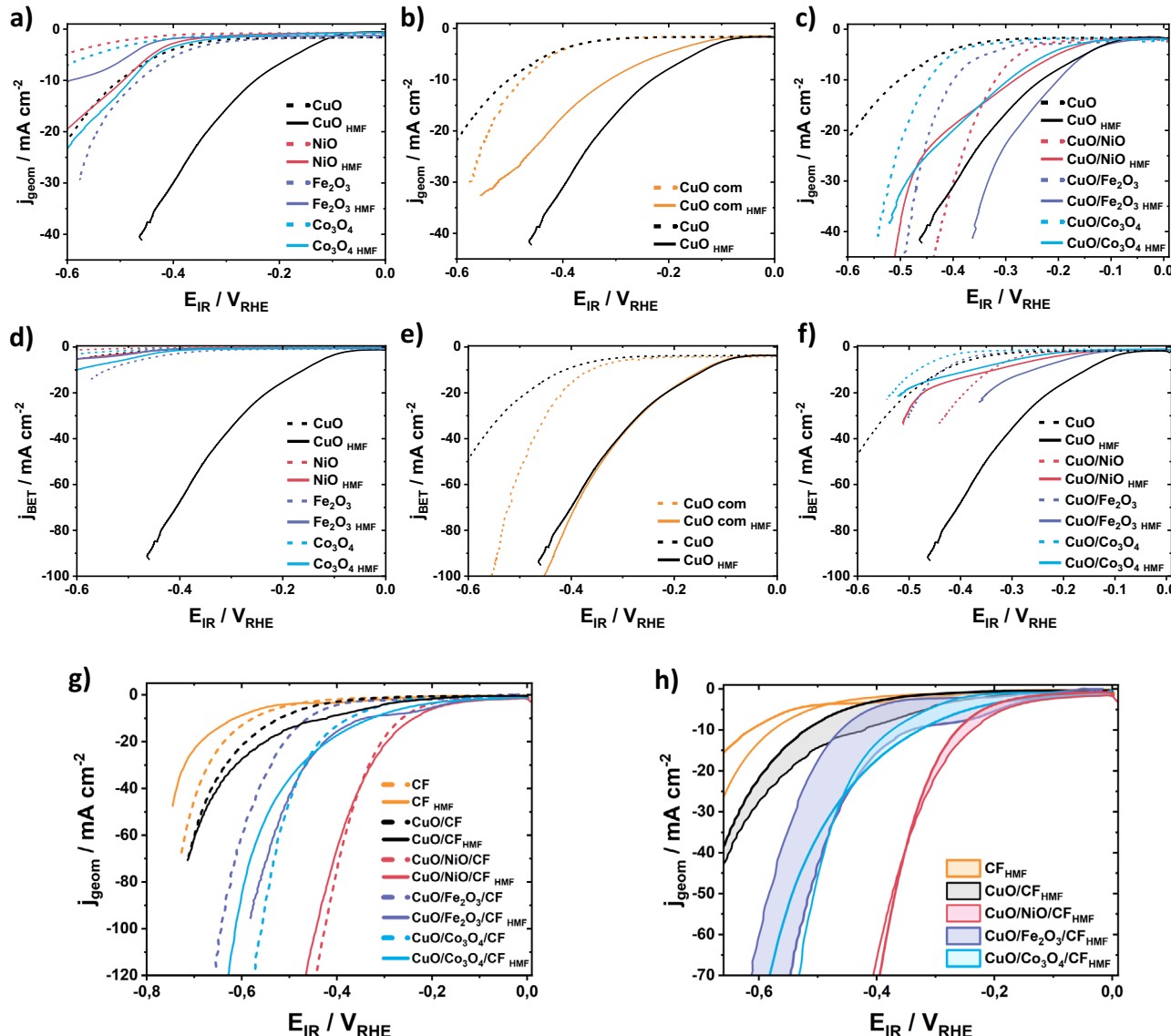

**Fig. 3 | RDE and undivided three-electrode cell measurements of CuO/MOₓ and CuO/MOₓ/CF.** a RDE measurements of pure CuO (black), NiO (red), Fe₂O₃ (purple), and Co₃O₄ (blue). **b** RDE measurements of commercial CuO (orange) and pure CuO (black). **c** RDE measurements of mixed metal oxides CuO/NiO (red), CuO/Fe₂O₃ (purple), and CuO/Co₃O₄ (blue). **d–f** BET corrected current densities of the corresponding plots from (**a–c**). All RDE LSV measurements were taken between 0 V_RHE to −0.6 V_RHE at a scan rate of 10 mV s⁻¹ in 0.1 M KOH with (solid line) and without (dashed line) 10 mM HMF at 2500 rpm with an electrode surface area of 0.19 cm² and a catalyst loading of 0.04 mg. All measurements are 100% manual internal

resistance (IR) corrected. **g** Undivided three-electrode cell (UTEC) measurements of CuO and CuO/MOₓ on CF with CF (orange), CuO/CF (black), CuO/NiO/CF (red), CuO/Fe₂O₃/CF (purple), and CuO/Co₃O₄/CF (blue). **h** Blow up of the UTEC measurements in (**g**) with colored HMFRR selectivities areas of CF (orange), CuO/CF (black), CuO/NiO/CF (red), CuO/Fe₂O₃/CF (purple), and CuO/Co₃O₄/CF (blue). All UTEC LSV measurements were taken between 0 V_RHE to −0.8 V_RHE at a scan rate of 10 mV s⁻¹ in 0.1 M KOH with (solid line) and without (dashed line) 10 mM HMF without rotation and an electrode area of 1 cm² and a catalyst loading of 1 mg cm⁻². All measurements are 100% manual internal resistance (IR) corrected.

We now turn to the catalytic HMFRR RDE performance of the two-phase oxide catalysts. The co-precipitated 10 mol% $MO_x$ phase was designed to enhance the HMFRR of the OD-Cu phase by possible Hydrogen spillover. Figure 3c indeed confirms that the secondary phase provided the desired increase in HER activity (dashed lines) in the order $CuO < CuO/Co_3O_4 < CuO/Fe_2O_3 < CuO/NiO$. More importantly, the HMFRR reactivity of all $CuO/MO_x$ catalysts substantially increased. The cathodic potential shifts at 10 mA cm⁻² between the HER and the HMFRR ranged at 200, 270, 37, and 150 mV for $CuO/Fe_2O_3$, CuO, CuO/NiO, and $CuO/Co_3O_4$, respectively. The absolute HMFRR performance at 10 mA cm⁻² and at −0.3 V_RHE dropped in that same order. The $CuO/Fe_2O_3$ catalysts outperformed the reference CuO catalysts and became one new catalyst of further interest. At increasing current densities and more negative electrode

potentials, CuO/NiO, and $CuO/Co_3O_4$ displayed cross-over points where the current density in presence of HMF equalled that in absence (pure HER). Prior to the cross-over point, the catalysts showed electrode potential ranges of primary interest for device operation with substantial HMFRR reactivity. Clearly, based on the vastly different BET surface areas between CuO (22 m²/g) and the $CuO/MO_x$ (69–88 m²/g), the corrected reactivity trends were $CuO > CuO/Fe_2O_3 > CuO/NiO ~ CuO/Co_3O_4$. We confirmed our hypothesis that modulation of the reactivity of CuO is possible by the presence of the second crystalline metal oxide phase (Fig. 2f–j and Supplementary Table 1). We can also add that our (co-)precipitation-calcination (air) synthesis protocol has slight catalytic advantages over the physical mixing of the individual precipitated metal oxides (Supplementary Fig. 14).

## Catalytic testing of stationary foam electrodes

We carried out a stepwise scale-up from the RDE powder thin film level to the 5 cm² spray-coated CF-supported cathodes level, which were to be deployed in alkaline exchange membrane (AEM) flow electrolyzer cells. To achieve this, we prepared a 1 cm² electrode by spray-coating CuO/$MO_x$ powders onto a metallic Cu foam (CF) support (Supplementary Fig. 13b), denoted CuO/$MO_x$/CF. We tested the spray-coated film electrodes in an undivided three-electrode cell configuration (UTEC) without rotation of the working electrode (Supplementary Fig. 15). The reactivity of the HMFRR rather than its chemical selectivity was of focus here. After a loading study (Supplementary Fig. 16) to select suitable catalyst loadings, a catalytic HMFRR activity screening with and without HMF (Fig. 3g, h) established the most favorable current-potential operating regions for the design of an efficient AEM HMF electrolyzer. HMFRR, as well as baseline HER currents of the spray-coated electrodes, increased substantially compared to the powder thin film RDE study. The HMFRR activity increased in the order CF < CuO/CF < CuO/Co₃O₄/CF ~ CuO/Fe₂O₃/CF < CuO/NiO/CF over the entire current range. The current density differences between baseline HER and HMFRR, however, dropped significantly to a value between 22 to 117 mV. Clearly, the background HER reactivity sharply increased in the sprayed thin film format. This is due to the larger surface area of the foam support and the thicker catalyst films. At the same time, however, Supplementary Fig. 17 shows that CF alone does not necessarily lead to a generally increased performance. Here it becomes clear once again that the combined CuO/$MO_x$ metal oxides with or without CF support bring an increase in activity.

## Alkaline exchange membrane HMF electrolyzer cell tests

Based on our CuO/$MO_x$/CF catalyst discovery and characterization studies above, we moved to build the first 5 cm² active area alkaline exchange membrane electrolyzer for the valorization of HMF in strongly alkaline pH on the cathode, coupled to the alkaline oxygen evolution reaction on the anode using all PGM free catalysts (Supplementary Fig. 18a). As the cell operated, we monitored the conversion of HMF ($X_{HMF}$), the selectivity of products such as BHMF and MFA ($S_{BHMF}$ and $S_{MFA}$) as well as the Faradaic efficiencies (FE) of H₂, BHMF, and MFA (Fig. 4). Other reaction products, such as Methylfurfural (MF) or di/polymerized HMF are referred to as others.

Figure 4a–e compares the faradaic product efficiencies of the uncoated CF support reference compared to the CuO/$MO_x$/CF electrodes at varying applied currents. Supplementary Fig. 19 and Supplementary Table 3 report the polarization behavior, the detailed time-resolved cell potentials during the galvanostatic step protocol, as well as the corresponding H₂ rates and performance parameters for all catalysts.

For the pure metallic Cu foam (CF) support, high $FE_{BHMF}$ and $FE_{others}$ were evident (Fig. 4a). $FE_{H2}$ and the absolute hydrogen production rate increase monotonically with current density and cell potential (Fig. 4a and Supplementary Fig. 19a). The 2e⁻ reduction of the −COH aldehyde group appears to be fast and fairly selective on metallic Cu surfaces. Note that no MFA was produced in Fig. 4a suggesting that the (2 + 2)e⁻ sequential hydrogenation/hydrogenolysis of HMF to MFA in high alkaline environment is unfavorable (Fig. 4a and Supplementary Fig. 13b).

The CuO/CF reference electrode catalyzed both 2e⁻ hydrogenation and some subsequent 2e⁻ hydrogenolysis to MFA in alkaline conditions (Fig. 4b and Supplementary Fig. 19b). The $FE_{MFA}$ values exceeded $FE_{BHMF}$ at all current densities and peaked at 20 mA cm⁻². Again, HER increases monotonically and appears to compete with MFA production.

The CuO/NiO/CF cathode displayed somewhat lower $FE_{MFA}$ values in favor of $FE_{HER}$, in agreement with the well-documented HER reactivity of Ni oxides in alkaline conditions (Fig. 4c and Supplementary Fig. 19c)[35]. The 20 mA cm⁻² appears as narrow MFA selectivity sweet spot, in line with the narrow selective operating regime of CuO/NiO/CF in Fig. 3h. We conclude that a high catalytic HER reactivity of the

secondary $MO_x$ oxide is in direct mechanistic competition to MFA production. We rationalize this as a competition for adsorbed H atoms, $H_{ad}$ needed in Langmuir-Hinshelwood-type reaction pathways to H₂ and MFA. We further hypothesize that BHMF, on the other hand, is largely formed in direct proton reduction according to an Eley-Rideal-type process. This is why BHMF production appears unaffected by varying HER production. Interestingly, low $FE_{others}$ values suggested an additional proton/$H_{ad}$ competition with di/polymeric HMF products with high proton demand.

The CuO/Fe₂O₃/CF cathode (Fig. 4d and Supplementary Fig. 19d) is the most interesting one. It displayed not only the lowest cell potentials (highest energy efficiencies) (Supplementary Fig. 19d) but most importantly, offered the most favorable $FE_{MFA}$ values at very low $FE_{others}$, $FE_{H2}$, and $FE_{BHMF}$, particularly at 10 mA cm⁻². CuO/Fe₂O₃/CF is an excellent MFA-producing hydrogenation/hydrogenolysis catalyst over larger current ranges, as anticipated in Fig. 3h. Again, FE values confirm the hydrogen competition between HER and MFA production, while BHMF formation is unaffected. CuO/Fe₂O₃/CF proves that with rising HER, the hydrogenation-hydrogenolysis-selectivity is developing in favor of hydrogenation.

The CuO/Co₃O₄/CF cathode (Fig. 4e and Supplementary Fig. 19e) displayed favorable $FE_{MFA}$ values up to 20 mA cm⁻², after which they dropped quickly as the $FE_{H2}$ increased. This is in excellent agreement with results from Fig. 3h, where HMFRR and HER polarizations cross around at current. The secondary Co₃O₄ catalyst leads to high $FE_{others}$ (Fig. 4e).

AEM HMF electrolyzer tests with a CuO/NiO/CF cathode were extended until complete HMF conversion, while the time-resolve evolution of $FE_{products}$ and $X_{HMF}$ (Fig. 4f) was tracked in the electrolyzer exit feed. Interestingly, MFA was selectively produced from HMF at a nearly 60% conversion during the first 15 min. At that point in time, MFA production through hydrogenation/hydrogenolysis suddenly ceased, while the hydrogen evolution dominated the electrolysis products at moderate BHMF production. In part, we attribute the sharply lower HMF to MFA conversion to depleted local HMF concentration at the surface associated with lower coverages of reactive HMF intermediates, such as BHMF. Sustained proton surface reduction and high $H_{ad}$ coverage render Volmer-Tafel HER the dominant process. We note that a compositional or structural change of the catalyst surface cannot be fully excluded as an origin for the FE variations, yet the available catalyst stability data (Supplementary Figs. 8–13) make this hypothesis unlikely. We also note that the HMFRR/HER competition at the surface is in congruence to earlier own work on the competition of oxidative HMF conversion and the oxidation oxygen evolution reaction[3]. We further conclude from the fact that no MFA was converted to 2.5-Dimethylfuran (DMF) nor BHMF was converted to MFA after 15 min, that the electrocatalytic reduction of −CH₂OH alcohol group is extremely unfavorable in alkaline conditions[10,11]. A partial deprotonation of −CH₂OH at pH 13 to less reactive alcoholate groups, −CH₂O⁻ (pKa ~ 12)[36,37] is compounding the kinetic hindrance.

To display the impact of the secondary oxide phase on the competition between hydrogenolysis and hydrogenation, in other words on the MFA/BHMF selectivity preference, we plotted the HMF conversion versus the absolute $S_{MFA}/S_{BHMF}$ ratio in Fig. 4g–i. As HMF conversions rise to completion ($X_{HMF} = 1$ for CuO/Fe₂O₃/CF at 20 and 30 mA cm⁻²), the $S_{MFA}/S_{BHMF}$ ratio follows the FE trend. Hydrogenation prevails over hydrogenolysis at higher currents and conversions. Only at 10 mA cm⁻² for CuO/Fe₂O₃/CF and CuO/Co₃O₄/CF, where no or hardly any hydrogen is formed, hydrogenolysis is preferred over hydrogenation.

## AEM HMF electrolyzer stability tests

We checked the galvanostatic stability of our AEM electrolyzers, for which we used the favorable CuO/Fe₂O₃/CF electrode at 20 mA cm⁻² (Fig. 5a) due to its favorable cell parameters, such as overpotential,

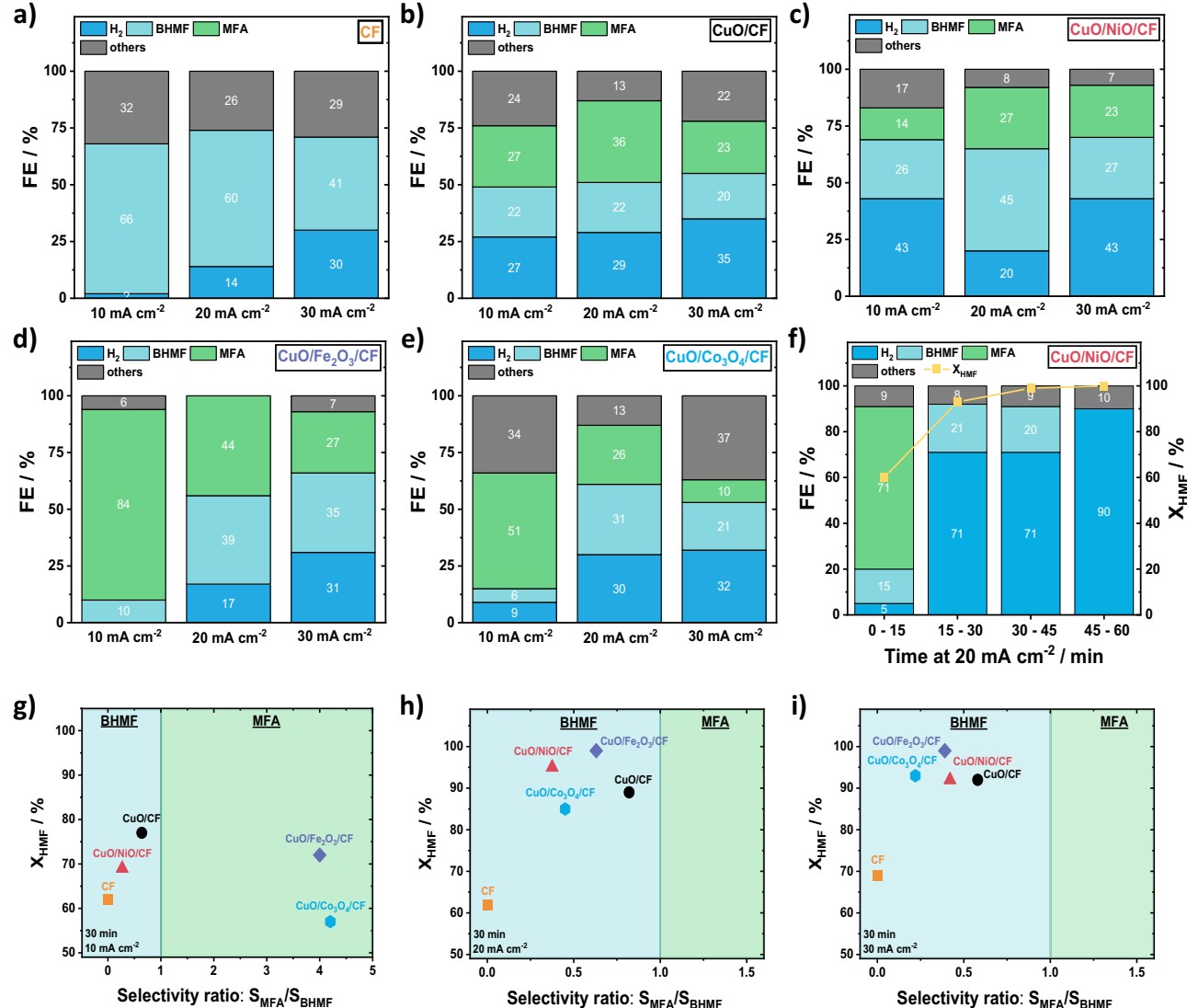

**Fig. 4 | MEA-flow-cell performance measurements of CuO/MO$_x$/CF electrodes.**
**a–e** Faradaic efficiencies for H$_2$ (strong blue), BHMF (light blue), MFA (green), and other products (gray) of the different spray-coated CuO/MO$_x$/CF catalysts.
**f** Faradaic efficiencies and HMF conversion (yellow) are calculated for every 15 min time interval over 60 min using CuO/NiO/CF as a catalyst. Product color code stays as in (**a–e**). **g–i** Scatter plot of the product selectivity preference. HMF conversion over MFA/BHMF selectivity ratio, calculated by S$_{MFA}$/S$_{BHMF}$ for all CuO/MO$_x$

catalysts on CF at different current densities for 30 min. The color code stays the same as before. Cell reaction conditions: 0.1 M KOH with 10 mM HMF as catholyte (100 ml, recycled), 0.1 M KOH as anolyte (100 ml, recycled), 5 cm$^2$ electrode area, nickel foam (NF) as anode and a flow rate of 25 ml min$^{-1}$, at 10–30 mA cm$^{-2}$ for 30 min. High frequency resistance results are between 0.7–1.05 Ω (Supplementary Fig. 18b). Relative errors of 2–4% for FE$_{products}$, 3–5% for FE$_{H2}$, and 1–3% for X$_{HMF}$ resulted (Supplementary Fig. 20).

HMF conversion, Faradaic efficiencies and selectivities (Fig. 4 and Supplementary Table 3). To validate CuO/Fe$_2$O$_3$/CF as a stable electrode, we tracked the conversion of HMF and the selectivities of BHMF and MFA over 5 test protocol cycles of 2.5 h. Figure 5a reveals 100% HMF conversion at constant high selectivities.

In the last electrolyzer design step, we replace our standard Nickel foam (NF) anode with a previously reported highly active NiFe(-Cl$^-$)-LDH@NF material, an excellent OER catalyst and HMF oxidation catalyst (Fig. 5b, HMF//KOH)[3,38]. We used the CuO/Fe$_2$O$_3$/CF//NiFe(-Cl$^-$)-LDH@NF cathode//anode AEM water electrolyzer (no HMF feed) as a reference (KOH//KOH in Fig. 5b). Compared to the AEM water electrolyzer cell, the CuO/Fe$_2$O$_3$/CF//(NiFe(-Cl$^-$)-LDH@NF) cathode//anode design with HMF//HMF demonstrated a 20% drop in cell input voltage at stable S$_{BHMF}$, S$_{MFA}$ and stable S$_{FDCA}$, the HMF oxidation product Furan Dicarboxylic Acid (FDCA). A closed 100% HMF reduction product balance remained elusive due to HMF polymerization (humin formation).

## Mechanistic discussion

Figure 1a displays the rich electrochemical reactivity of the HMF molecule. The 2e$^-$/2H$^+$ reduction of a −CH$_2$OH group (of HMF or BHMF) into a −CH$_3$ group is associated with a C-O bond breaking and, as such, is likely to require a surface-adsorbed reactive state of the HMF intermediate and adsorbed H$_{ad}$ in atomic proximity, following a Langmuir-Hinshelwood-type reaction. By contrast, it appears feasible that the simpler 2e$^-$/2H$^+$ reduction of the aldehyde group −COH to the −CH$_2$OH group (no bond breaking, mere bond order reduction) is catalyzed using protons from the double layer rather than H$_{ad}$, following the Eley-Rideal pathway. Indeed, on CuO catalysts with their rough metallic OD-Cu surface state, moderate MFA yields were detected. Undercoordinated Cu adatoms on the roughened Cu surface facets activate -COH groups of (B)HMF molecules. The secondary MO$_x$ phase, remaining an oxidic surface state, offers a balanced hydrogen chemisorption to H$_{ad}$, and, as such, may act as a source or pool of reductive H$_{ad}$ equivalents that diffusively spill over and aid in the

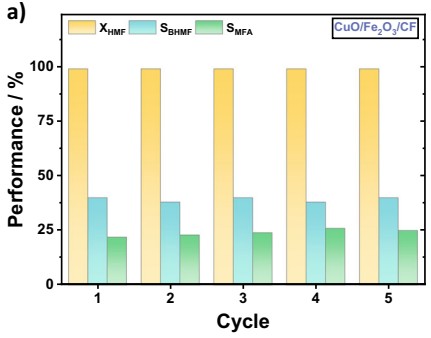

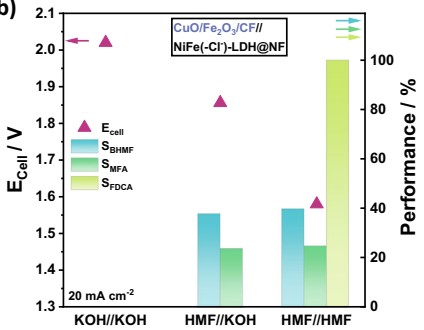

**Fig. 5 | MEA-flow-cell stability measurements of CuO/Fe₂O₃/CF and combination of HMF reduction and oxidation. a** Stability measurements of CuO/Fe₂O₃/CF over five cycles (2.5 h) at 20 mA cm⁻². HMF conversion in yellow, BHMF selectivity in turquoise, and MFA selectivity in green. **b** cell potential and performance comparison between non-HMF containing electrolytes on both sides, HMF at the cathode and HMF on anode and cathode, using CuO/Fe₂O₃/CF as the cathode and NiFe(-Cl⁻)-LDH@NF as the anode. The cell potential is given in pink triangles, BHMF selectivity in turquoise, MFA selectivity in green, and FDCA selectivity in light green. Cell reaction conditions: 5 cm² electrode area, flow rate of 25 ml min⁻¹, at 20 mA cm⁻² for 30 min.

stepwise reduction of adsorbed HMF molecules and its derivatives (Fig. 1b). HER and MFA production compete for $H_{ad}$.

Acidic electrolyte conditions suppress undesired electroless di- and polymerization of HMF molecules, yet require noble and corrosion stable catalysts. Alkaline conditions favor humin formation, however enable cost-effective catalysts. Under the strongly alkaline conditions of the present study (pH 13) some of the alcoholic –CH₂OH protons (pKa ~ 12) are deprotonated forming negatively charged alcoholate groups, –CH₂O⁻, which are no longer available for hydrogenolytic reduction to methyl groups of MFA. Hence, MFA formation becomes more difficult under alkaline conditions[10]. We suppose, if reactive BHMF intermediates, however, remain adsorbed on the surface after the 2e⁻/2H⁺ aldehyde reduction, and are supplied with additional reducing $H_{ad}$ equivalents on the surface, this scenario favors the subsequent 2e⁻/2H⁺ hydrogenolysis to MFA.

We have explored the electrocatalytic reduction and valorization of HMF on novel noble metal-free Cu-based two-phase oxide precursor catalysts. We have reported a new noble metal-free oxide catalyst and the first electrochemical interface inside an HMF AEMWE that catalyzes the electrochemical HMF reduction to MFA in strongly alkaline conditions. Of particular focus was the competition between the 2e⁻/2H⁺ –COH hydrogenation to –CH₂OH and its subsequent 2e⁻/2H⁺ hydrogenolysis to –CH₃. We hypothesized that the nature of the secondary metal oxide and its HER activity will tune the HMF reduction selectivity of CuO via its own HER reactivity. That hypothesis that we validated. The use of alkaline electrolyzer conditions has practical implications, as the use of cost-effective catalysts and the coupling of the HMF reduction process to alternative anode reactions become possible. This was demonstrated by building a paired HMF-HMF electrolyzer.

## Methods

### Precipitation synthesis of pure metal oxides (MO$_x$; M = Cu, Ni, Fe, Co)

Preparation of the pure metal oxides (MO$_x$; M = Cu, Ni, Fe, Co) was carried out by precipitation method. In total, 200 mg of powdered metal salt precursor (Cu(II)Cl₂, Sigma-Aldrich 99%; Ni(II)(NO₃)₂ 6H₂O, Roth 99%; Fe(II)Cl₂ 4H₂O, Alfa Aesar 98%; Co(II)(NO₃)₂ 6H₂O, Acros Organics 99%) was solved in 27 ml of water. NaOH (Sigma-Aldrich 99.99% trace metals basis, 3 M, 3 ml) was added dropwise to the mixture. The precipitate was collected by centrifuge and was washed three times with H₂O, ETOH, H₂O and afterward dried by lyophilization. Then, obtained powder was calcined at 300 °C in a muffle furnace (Carbolite) for 3 h with a heating rate of 5 °C/min.

### Precipitation synthesis of mixed metal oxides (CuO/MO$_x$ 10 mol%; M = Ni, Fe, Co)

Preparation of the mixed metal oxides (CuO/MO$_x$; M = Ni, Fe, Co) was carried out by precipitation method. To achieve a molar ratio of 9/1 Cu/M, 116 mg of powdered Cu metal salt precursor (Cu(II)Cl₂, Sigma-Aldrich 99%) was mixed with 18–26 mg of second metal salt precursor (Ni(II)(NO₃)₂ 6H₂O, Roth 99%; Fe(II)Cl₂ 4H₂O, Alfa Aesar 98%; Co(II)(NO₃)₂ 6H₂O, Acros Organics 99%) and solved in 17 ml of water. NaOH (Sigma-Aldrich 99.99% trace metals basis, 3 M, 3 ml) was added dropwise to the mixture. The precipitate was collected by centrifuge and was washed three times with H₂O, ETOH, H₂O and afterward dried by lyophilization. Then, obtained powder was calcined at 300 °C in a muffle furnace (Carbolite) for 3 h with a heating rate of 5 °C/min.

### Physicochemical characterization of the prepared catalysts

Scanning electron microscope and elemental mapping (SEM-EDX) measurements were carried out with Zeiss Gemini 982 instrument.

Powder X-ray diffraction was performed using a Bruker D8 Advance apparatus. A Cu Kα (1.54 Å) radiation source was used. The wide-angle measurements were taken in the range of 5°–80°.

Thin film X-ray diffraction was performed using a Bruker D8 Advance apparatus. A Cu Kα (1.54 Å) radiation source was used. The wide-angle measurements were taken in the range of 10°–80°.

Transmission electron microscope (TEM) measurements were conducted using a Tecnai G2 20 s-Twin microscope, equipped with a LaB6-cathode and a GATAN MS794 P CCD-detector at ZELMI Centrum, Technical University of Berlin. TEM samples were ultrasonicated in i-PrOH and drop-dried on copper grids.

X-ray photoelectron spectroscopy (XPS) measurements were carried out with a Thermo Scientific K-Alpha+ X-ray photoelectron spectrometer (Group of Prof. Thomas, TU Berlin). The measured spectra were analyzed using CasaXPS software. Binding energy (BE) was aligned by the C 1s spectra.

The surface areas of the metal oxides and mixed metal oxides were investigated by N₂ physisorption with Autosorb-1 (Quantachrome Instruments). A 4 mm diameter glass tube was stacked with the sample, glass wool, and a glass rod for minimizing the dead volume. The sample weight was adjusted so that the absolute surface area exceeds 10 m². To remove all gas and water residues the sample was degassed under vacuum at 80 °C for at least 24 h. The adsorption and desorption isotherms were recorded in a range of $10^{-5} \leq p/p_0 \leq 0.995$ with $p_0$ referring to the saturation pressure and $p$ the actual gas pressure.

### Inductively coupled plasma optical emission spectroscopy (ICP-OES)

ICP-OES was carried out with a Varian 715-ES instrument. All samples were prepared by diluting 1 mg of catalyst in Aqua Regia (1 ml) and ultrapure water (9 ml).

## Rotating disk electrode (RDE) measurements and procedure

The three-electrode RDE measurements were carried out by using a BioLogic Sp-200 potentiostat. It was measured against a reversible hydrogen electrode (RHE) as the reference electrode and a Pt-mesh as the counter electrode. The HER and 5-HMF reduction activities were investigated in 50 ml $N_2$-saturated 0.1 M KOH with and without 10 mM 5-HMF at room temperature (RT). Drop casted catalyst powder on a glassy carbon disc electrode with an electrode area of 0.196 cm$^2$ and a catalyst loading of 0.04 mg at 2500 rpm was used as the working electrode. For cyclic voltammetry (CV) scan rates between 5 mV s$^{-1}$ and 50 mV s$^{-1}$ in a potential range between 0 V and −0.6 V (if not noted otherwise) were realized.

## Undivided three-electrode cell measurements and procedure

The three-electrode electrochemical measurements of the $1 \times 1$ cm prepared cathodes (1 mg cm$^{-2}$) were carried out using a BioLogic Sp-300 potentiostat. It was measured against a reversible hydrogen electrode (RHE) as the reference electrode and a Pt-mesh as the counter electrode. The HER and 5-HMF reduction activities were investigated in 25 ml $N_2$-saturated 0.1 M KOH with and without 10 mM 5-HMF at room temperature (RT). For cyclic voltammetry (CV) scan rates between 5 and 50 mV s$^{-1}$ in a potential range between 0 and −0.8 V were realized. Spray-coating technique was used to prepare the cathode with an aimed loading of 1 mg cm$^{-2}$. The catalyst ink was prepared by mixing 100 µl Mili-Q water, 5 ml i-PrOH and 100 mg catalyst. During sonification ionomer solution (3 wt% Nafion©-solution, 1100 g/mol, Sigma-Aldrich) was added, resulting in a 1 wt% dispersion of the catalytic active material. Due to the static character of the working electrode (WE) holder, no rotation was applied.

## Full-cell measurements and fabrication of the applied electrodes

The 5 cm$^2$ MEA-type cell measurements were conducted using a commercial cell from Dioxide Materials. Linear serpentine flow fields made out of $TiO_2$ (anode and cathode) and an electrolyte flow of 25 ml min$^{-1}$ was applied. 0.1 M KOH with and without 10 mM 5-HMF was used as the electrolyte. Blank Nickel Foam (NF), as well as the prepared and modified NiFe(-Cl$^-$)-LDH@NF electrode serve as anodes. Furthermore, spray-coating technique was used to prepare the cathode with an aimed loading of 1 mg cm$^{-2}$. The catalyst ink was prepared by mixing 100 µl Mili-Q water, 5 ml i-PrOH and 100 mg catalyst. During sonification ionomer solution (3 wt% Nafion©-solution, 1100 g/mol, Sigma-Aldrich) was added, resulting in a 1 wt% dispersion of the catalytic active material. The measurements were conducted in 0.1 M KOH (99.99%, Sigma-Aldrich). System activation was realized with 20 CVs from 0−−0.6 $V_{cell}$ at a scan rate of 50 mV s$^{-1}$. At constant current densities of −10, −20, and −30 mA cm$^{-2}$ (except noted otherwise) HMF electrolysis was carried out over 30 min, close to the theoretical charge needed for full HMF conversion to BHMF. Potentio electrochemical impedance spectroscopy was measured before and after constant current electrolysis between 100,000−1000 Hz. The stability tests were carried out under the same conditions at −20 mA cm$^{-2}$ for 5 cycles in 0.1 M KOH and 10 mM 5-HMF.

## Product analysis

Product analysis was carried out before, during, and after a constant applied current density. The samples (500 µl) were taken by an automatic aliquot sampling device and analyzed by high performance liquid chromatography (HPLC, Agilent 1200 series, Agilent Technologies Zorbax SB-C18 column) at a constant flow rate of 0.8 ml min$^{-1}$ ($H_2O$:$C_2H_3N$ 95:5) and detected by a refractive index detector (RID).

Hydrogen was quantified by an Unisense© Hydrogen Sensor. The sensor was directly included in the cathodic outlet line via a T-piece. Calibration was done in 0.1 M KOH saturated with $H_2$/Ar 5/95 gas mixture.

## Calculation of HMF conversion, oxidation product yields and Faradaic efficiency

$$X_{\text{HMF}} = \left(1 - \frac{n_{\text{HMF,remaining}}}{n_{\text{HMF,initial}}}\right) \times 100\% \qquad (1)$$

$$Y_x = \frac{n_x}{n_{\text{HMF,initial}}} \times 100\% \qquad (2)$$

$$\text{FE}_x = \frac{n_x}{\frac{Q_{\text{total}}}{F \times z}} \times 100\% \qquad (3)$$

## Data availability

The data generated in this study are provided in Supplementary Information.

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

## Acknowledgements

The research leading to these results has received funding from the European Union's Horizon 2020 research and innovation program under grant agreement no. 851441, SELECTCO2 and grant agreement no. 101006701, ECOFUEL. Financial support by the German Research Foundation (DFG) through Grant Reference Number STR 596/12-1, the Federal Ministry for Education, Research and Development (Bundesministerium für Bildung und Forschung, BMBF) under Grant numbers 03SF0613D and 03SF0611A in the collaborative research projects AEMready and H2-Meer and funding by European Union's HORIZION.3.1—The European Innovation Council (EIC) Programme through the grant agreement 101071111—ANEMEL are gratefully acknowledged. XPS measurements were performed by the Group of Prof. Arne Thomas, TU Berlin HR-STEM measurements were performed by Zentraleinrichtung Elektronenmikroskopie (ZELMI), TU Berlin.

## Author contributions

Design and execution of the study: P.H.; SEM images: T.M.; TEM images: M.K.; Wrote the manuscript: P.H. and P.S.

## Funding

## Competing interests

The authors declare no competing interests.
