## [Peer Review File · Nature Communications]

REVIEWER COMMENTS

Reviewer #1 (Remarks to the Author):

Biomass conversion represents a possible means to reduce dependence on these fossil fuels in favor of more environmentally benign and renewable syntheses of fuels and industrially useful compounds. Controlling hydrogenation and hydrogenolysis of HMF is critical to increasing the yield and selectivity of the desired product. This manuscript described the reduction of HMF on oxide-derived Cu bimetallic under alkaline conditions. They verified that the valorization of HMF to MFA via a four-electron process is possible. This result is very important for the fundamental understanding of reaction mechanism, as well as the advanced electrocatalysts design and construct. This manuscript can be accepted after major revision. The detailed comments are listed below:

1. The mass loading is essential for electrochemical performance. In order to give better comparison, the electrochemical activity in mass loading form should also present.
2. The ascription of XPS peak should be marked in Figure 2h. The Ni 2p_{3/2} consists two characteristic peaks, while only one peak in Ni 2p_{1/2}. The detailed explanation should provide.
3. The electrochemical activity of different electrocatalysts in BET surface area should provide.
4. More evidence for mechanism discussion should provide, rather than a proposed reaction pathway. For example, how the MO_x phase tune the catalytic selectivity?
5. Pt mesh counter electrode would give significant contribution for electrochemical activity of cathode, which will gradually deposit on cathode during electrochemical testing. The influence of counter electrode should be considered.
6. The manuscript should be carefully checked to make it clear to the readers. For example, 'Figure 1k' should be 'Figure 2k'; The details information for reference are incomplete;

Reviewer #2 (Remarks to the Author):

In this work, the authors report an oxide-derived Cu-based two-phase catalysts oxide-derived Cu/MO_x and employ it to electro-catalyze 5-hydroxymethylfurfural (5-HMF) valorization. The focus of this work is to demonstrate the nature of the second crystalline metal oxide phase (MO_x) on its HER activity and the HMF selectivity over CuO based catalyst. However, their interpretation is speculative and does not provide a clear understanding of the reasons for the various selectivity. For these reasons and details below, the present manuscript does not meet in my view the publication criteria of Nature Communication. Please find below some major points of concern:

- (1) In the Abstract, the abbreviation of OD-Cu catalyst appears for the first time. It is necessary to explain what it refers to.
- (2) In the Catalyst Synthesis section, the authors indicated that “the choice of a two- phase catalyst concept originated in the basic favorable HMFRR reactivity of pure crystalline CuO that we intended to tune by the nm-scale (rather than Angstrom-scale) presence a distinct oxide phase with varying structure and chemisorption characteristic.” Literatures should be cited here to support this conclusion.
- (3) The authors chose first row transition metals as the M element in MO_x, the descriptions on the reason why specific MO_x were chosen to perform the reaction especially in alkaline condition, including their respective functions are suggested to add in Introduction part.
- (4) For the figure 2 in page 7, since the Cu⁰ and Cu⁺ could be distinguished in Cu 2p XPS spectra, Cu LMM is suggested to be supplied.
- (5) In line 164-165, the authors claim that “...a very rough Cu surface characterize by many undercoordinated Cu adatoms which likely serve as active sites for activation steps of reactant molecules...”, Cu K-edge EXAFS is suggested to be added to acquire relevant local coordination information.
- (6) The BET results are quite different over series catalysts, then how the factor about specific surface area on catalytic performance be excluded. In other word, how is the size effect of CuO excluded? Otherwise, the role of surface catalysis in MO_x is too speculative without excluding these factors.
- (7) For the figure 2 in Page 11, the error bars are lacked. In theory, the experiments should be repeated at least 3 times to be conformable with presented results.
- (8) The mechanistic studies are very speculative and lack of support by either experiments or literatures (even if they are for thermocatalytic reaction, some information can be inferred). Particularly the C-OH group of the BAMF could be directly adsorbed into the MO_x, this is not considered in this work but likely an important path. The authors focus the competition between the 2e-/2H+ -COH hydrogenation to -CH₂OH and its subsequent 2e-/2H+ hydrogenolysis to -CH₃, it is better to perform DFT calculation aiming to provide more convincing evidence. In addition, series in-situ experiments such as in situ FTIR should be designed to investigate the adsorption/activation behavior of substrate or intermediated products molecules.
- (9) How about the catalytic performance in comparison with peer work especially with other non-noble catalysts. To highlight the performance advantages of OD-Cu/MO_x, the evaluation results of single factors such as substrate conversion, product selectivity and other reaction condition especially temperature should be expressed through a radar graph in SI.
- (10) In order to further determine the advantages of the two-phase catalyst, it is recommended to assess the performance of the physical mixture of two single oxides for comparison. This necessary contrast is indispensable.
- (11) The authors claim that such two-phase structure is induced during the reaction process. The specific structural dynamic evolution needs to be clearer.
- (12) How about the stability of OD-Cu/MO_x catalysts. Relevant reusability test should be added.

(13) The authors mentioned “Alkaline conditions favor humin formation”, but did not give the selectivity of di- and polymerization products. Have the di- and polymerization products been detected and have the carbon balance been calculated in this work?

Reviewer #3 (Remarks to the Author):

Recommendation: Minor revisions needed

Comments:

The topic of this manuscript is relevant for Nature Communications. The manuscript is well written, the characterization methodology is sound, the authors provided enough detail to reproduce their work, and the results are very important for the electrochemical community. However, it contains a few experimental and analytical issues that the authors need to address before it can be accepted for publications. Therefore, I recommend reconsidering after doing minor revisions.

In summary, the authors explored the electrochemical reduction of 5-Hydroxymethylfurfural (HMF) to 5-Methylfurfuryl alcohol (MFA) under strongly alkaline reaction environments over oxide-derived Cu bimetallic electrocatalysts. The authors investigated the relationship between the surface catalysis CuO on NiO, Fe₂O₄, Co₃O₄ surfaces and the selectivity towards different reduction products such as 2,5-Bishydroxymethylfuran (BHMF) and hydrogen evolution reaction (HER). The authors provide evidence for a kinetic competition between the HER and hydrogenolysis of BHMF to MFA and showed that the product selectivity depends on the electrode composition and current density. However, each material had different (electrochemical) surface area (Figure 2K) and required different potentials to generate the same current density (Figure 3) when tested in Rotating Disk Electrodes (RDE) and batch cells; hence, the comparison should not be done as a function of current density but as a function of potential.

Additionally, all the electrodes showed similar electrochemical performance (but different product selectivity) when tested in the alkaline exchange membranes (AEM) flow cell. The authors did not measure or report the half-cell potential in the flow cell so this discrepancy in electrochemical performance between the flow and batch systems cannot be explained. The authors are encouraged to compare the performances at equal IR-compensated potentials between the batch and flow system to better understand the results.

For this reason, this manuscript cannot be accepted as it is, and the authors are encouraged to address the following points:

1. The authors show the XPS spectra of CuO, NiO, Fe₂O₃, and Co₃O₄. Do the authors observe any change in oxidation state after reaction?
2. The performance of CuO/NiO/CF in Figure 3 g and f is very impressive, but it is not clear if the boost in performance comes from the CuO/NiO or NiO/CF. The authors are recommended to have a plot comparing the performances with and without HMF of CuO/CF, CuO/NiO, NiO/CF, and CuO/NiO/CF. Same for the Fe₂O₃/CF and Co₃O₄/CF systems.
3. What percentage of IR correction did the authors apply?
4. The normalization based on BET can be misleading as not all the area is electrochemically active. The authors are recommended to obtain the real electroreactive surface area (ECSA) based on capacitance. If that is not available, the authors are recommended to just use the BET of the active phase, CuO, to evaluate the trends.
5. The authors report the electrochemical performances (cell potentials and H₂ rates) of the MEA-flow cell for the different systems in Figure S16; that is a) CF, b) CuO/CF, c) CuO/NiO/CF, d) CuO/Fe₃O₄/CF, and e) CuO/Co₃O₄/CF. The authors show that CF at 10 mA/cm² nearly identical activity as the other materials. Similar operation voltages were observed at 20 mA/cm² for all the systems as well. At 30 mA/cm², all the systems appeared to have better performance (more stable cell voltages) than the CF itself. Hence, it is unclear what the CuO/MxO_y is doing at 10 and 20 mA/cm², and the authors are not encouraged to draw conclusions from these experiments.
6. The fact that the RDE and batch experiments (Fig 3) systems gave such different performances for the different electrodes evaluated while the experiments in flow cell had similar performances suggest that the flow system is not kinetically limited, and the performance might be controlled by different effects such as IR or mass transfer. The authors should measure the half-cell potential in the flow cell to better correlate the performance of the flow system with that of the RDE and batch systems.
7. In Figure 4, the authors are comparing the electrochemical performances at different current densities in the flow MEA cell but Figure S16 show that all the electrodes operated at nearly identical conditions (full cell voltage and H₂ rate with and without HMF). Without knowing the half-cell potential, it is difficult to rationalize whether the changes in product selectivity (Fig 4) are caused by differences in (i) potential as previously reported (e.g., ACS Catal. 2019, 9, 9964-9972 and ACS Sustainable Chem. Eng. 2020, 8, 4407-4418) or (ii) electrode composition.

8. It is unclear how the authors can conclude that “CuO/Fe₂O₃/CF showed potential for photo-electrochemical applications with maximum currents of up to 10 mA” as the authors never explored photo-electrochemical activities in this work nor mentioned it anywhere else in the manuscript. I think this line should be removed from the conclusion section.

9. Figure S1 appear to contain the same information as Figure 1. I think Figure S1 should be deleted.

10. Typos:

- Line 105. “:” should be a “.”
- Line 219: should “anodic” be changed to “cathodic”?
- Line 235: should “cm-2” be “cm²”
- Line 235: Cu foam already defined in line 233
- Line 250 should say Figure 3 instead of Figure 1. It is fine in the word doc but not in the pdf doc
- Line 332 should say Figure 4 instead of 2. It is fine in the word doc but not in the pdf doc.
- Line 355 should say Figure 5 instead of 3. It is fine in the word doc but not in the pdf doc.
- Line 386 replace “Alkaline Membrane HMF Electrolyzer” for “AME”

Rebuttal letter – Responses to Reviewers' comments

First and foremost, we would like to thank all reviewers for their constructive comments and intensive reviews.

Reviewer 1:

Biomass conversion represents a possible means to reduce dependence on these fossil fuels in favor of more environmentally benign and renewable syntheses of fuels and industrially useful compounds. Controlling hydrogenation and hydrogenolysis of HMF is critical to increasing the yield and selectivity of the desired product. This manuscript described the reduction of HMF on oxide-derived Cu bimetallic under alkaline conditions. They verified that the valorization of HMF to MFA via a four-electron process is possible. This result is very important for the fundamental understanding of reaction mechanism, as well as the advanced electrocatalysts design and construct. This manuscript can be accepted after major revision. The detailed comments are listed below:

1. The mass loading is essential for electrochemical performance. In order to give better comparison, the electrochemical activity in mass loading form should also present.

We agree with the reviewer that the mass loading is an essential tool in electrochemistry. As all loadings were 1 mg cm^{-2} for each prepared electrode, we divide by 1, which is why the values and trends don't change, just their unit.

To satisfy the reviewer's request we have now generated and inserted a mass loading corrected plot (Figure S16b) for the loading study (Figure S16a). Figure S16a shows the uncorrected data and Figure S16b the corrected data. It is clear that after correction all overpotentials converge. It is also clear that the loading of 1 mg cm^{-2} shows the best activity, which confirms our previous approach.

Figure S16: Undivided three-electrode cell (UTEC) loading study of CuO/NiO/CF. a) Showing the overpotential at 10 mA cm^{-2} for different catalyst loadings without (black) and with (green) 10 mM HMF. b) Showing the overpotential at 10 mA cm^{-2} (mass corrected) for different catalyst loadings without (black) and with (green) 10 mM HMF. The loading of 0 mg cm^{-2} is out of the plotted range.

2. The ascription of XPS peak should be marked in Figure 2h. The Ni $2p_{3/2}$ consists two characteristic peaks, while only one peak in Ni $2p_{1/2}$. The detailed explanation should provide.

We thank the reviewer for this comment. It is true that only the Ni $2p_{3/2}$ peak consists of two characteristic signals. The publication of Biesinger *et al.* for example, indicates that our Ni $2p_{3/2}$ signal

cannot only come from NiO, but must also have formed weak fractions of γ -NiOOH on the CuO/NiO powder surface.¹ We assume that γ -NiOOH was formed during the synthesis in the strongly alkaline milieu and that traces of it were not further oxidized to NiO. Syntheses leading to γ -NiOOH support this hypothesis.^{2,3} However, both the XPS data (**Figure 2h**) and the XRD data (**Figure 2b**) prove a dominant NiO structure.

We have rewritten and expanded the text to address this circumstance more clearly (**Main manuscript Page 5**).

*In addition, the characteristic peaks for NiO (872.6 eV and 855.3 eV), oxidic Fe (723.9 eV and 711.4 eV), oxidic Co (795.2 eV and 779.7 eV) as well as Ni²⁺ (853.7 eV and 872.6 eV), Fe³⁺ (732.1 eV and 718.7 eV) and Co^{2+/3+} (795.6 eV, 780 eV and 782.2 eV) are observed (**Figure 2h-j** and **Figure S3-S5**).^{1,4-7} Besides, for the Ni 2p_{3/2} signal (855.3 eV), γ -NiOOH was detected proportionally, which probably arose during the strongly alkaline synthesis and was not further oxidized to NiO (**Figure S3c**).¹⁻³*

In order to maintain clarity and uniformity in **Figure 2**, we decided to push the plot of individual signals into the SI (**Figure S3**) and hope to still act in the sense of the reviewer.

3. The electrochemical activity of different electrocatalysts in BET surface area should provide.

The BET corrected activities are given in **Figure 3d-f** for both MO_x and CuO/MO_x.

4. More evidence for mechanism discussion should provide, rather than a proposed reaction pathway. For example, how the MO_x phase tune the catalytic selectivity?

To improve our understanding of the mechanistic roles of the individual catalyst components CuO, MO_x and CF on the activity, further measurements were added and compared with the existing ones (**Figure S17**). The HER and HMFRR polarization curves of CuO/MO_x, CuO/CF, MO_x/CF and CuO/MO_x/CF were plotted for all MO_x. In particular, comparatively low HER and HMFRR activity can be seen for NiO/CF and Fe₂O₃/CF (**Figure S17** a and b). For Co₃O₄/CF this is also visible, but not quite as strong (**Figure S17c**). Furthermore, all CuO/MO_x/CF as well as the bimetallic powders CuO/MO_x are more active than pure CuO on CF. These two observations suggest a synergistic effect of the combined metal/ metal oxides.

Figure S 17: Activity comparison of CuO/MO_x, CuO/CF, MO_x/CF and CuO/MO_x/CF. comparing a) CuO/NiO (red), CuO/CF (black), NiO/CF (light red) and CuO/NiO/CF (dark red), b) CuO/Fe₂O₃ (purple), CuO/CF (black), Fe₂O₃/CF (light purple) and CuO/ Fe₂O₃/CF (shiny purple) and c) CuO/Co₃O₄ (blue), CuO/CF (black), Co₃O₄/CF (light blue) and CuO/Co₃O₄/CF (dark blue). Reaction conditions are the same as in Figure 3 for RDE and UTEC measurements.

5. Pt mesh counter electrode would give significant contribution for electrochemical activity of cathode, which will gradually deposit on cathode during electrochemical testing. The influence of counter electrode should be considered.

We thank the reviewer for the important objection. Platinum would indeed have a very positive effect on the HER activity and thus a very negative effect on our HMFRR activity and should therefore not participate in the reaction at the cathode.

However, the use of a platinum counter electrode is suitable for our RDE and UTEC systems or preferred over others for the following reasons.

1. The Pourbaix diagram shows that at pH=13 and a potential between 0 to -0.8 V, Pt is stable.⁸
2. The platinum counter electrode is only used for the activity tests in the RDE and UTEC. Thus, the reaction time is quite short and the leaching of platinum if there is some would be very low as a result. In addition, the electrolyte is replaced after each measurement.
3. When using a carbon-based electrode, it is very likely that this will reduce our product selectivity in favor of BHH.⁹

Despite all this, we have performed measurements for CF with a Pt and a C counter electrode (**Figure R 1**). These show that the Pt counter electrode does not improve the catalytic activity compared to a C counter electrode. Rather, the use of a C counter electrode shows the expected positive effect on the HMFRR (green solid line).

Figure R 1: UTEC measurements comparing CF with a Pt and a C counter electrode. Undivided three-electrode cell (UTEc) measurements of CF with Pt counter electrode (orange) and C counter electrode (green). UTEc LSV measurements were taken between 0 V_{RHE} to -0,8 V_{RHE} at a scan rate of 10mV s^{-1} in 0.1 M KOH with (solid line) and without (dashed line) 10 mM HMF without rotation and an electrode area of 1 cm^2 . All measurements are 100% manual internal resistance (IR) corrected.

6. The manuscript should be carefully checked to make it clear to the readers. For example, ‘Figure 1k’ should be ‘Figure 2k’; The details information for reference are incomplete.

We thank the reviewer for this comment and have reviewed the entire manuscript and corrected existing typographical errors.

Reviewer 2:

In this work, the authors report an oxide-derived Cu-based two-phase catalysts oxide-derived Cu/MO_x and employ it to electro-catalyze 5-hydroxymethylfurfural (5-HMF) valorization. The focus of this work is to demonstrate the nature of the second crystalline metal oxide phase (MO_x) on its HER activity and the HMF selectivity over CuO based catalyst. However, their interpretation is speculative and does not provide a clear understanding of the reasons for the various selectivity. For these reasons and details below, the present manuscript does not meet in my view the publication criteria of Nature Communication. Please find below some major points of concern:

(1) In the Abstract, the abbreviation of OD-Cu catalyst appears for the first time. It is necessary to explain what it refers to.

We thank the reviewer for the important comment and have defined OD-Cu in the abstract at the beginning

We demonstrate the first successful reduction of HMF to 5-Methylfurfuryl alcohol (MFA) under strongly alkaline reaction environments over oxide-derived Cu (OD-Cu) bimetallic electrocatalysts. (Main manuscript, Page 2)

(2) In the Catalyst Synthesis section, the authors indicated that “the choice of a two- phase catalyst concept originated in the basic favorable HMFRR reactivity of pure crystalline CuO that we intended to tune by the nm-scale (rather than Angstrom-scale) presence a distinct oxide phase with varying structure and chemisorption characteristic.” Literatures should be cited here to support this conclusion.

We agree with the reviewer and have added literature on tuning the activity of CuO by adding second metal oxides or changing the structure of the catalyst and thus the properties.

The choice of a two-phase catalyst concept originated from the good HMFRR reactivity of pure crystalline CuO that we intended to tune by the presence of a distinct second oxide phase at nm scale proximity (rather than by forming a new mixed oxide phase) with varying structure and chemisorption characteristics.¹⁰⁻¹³ (Main manuscript, Page 5)

Of course, a reference that directly addresses such activity tuning of CuO/MO_x in alkaline electrolyte for the HMFRR cannot be given, since we first investigated this. That’s why at the beginning of our study we just “intended”.

(3) The authors chose first row transition metals as the M element in MO_x, the descriptions on the reason why specific MO_x were chosen to perform the reaction especially in alkaline condition, including their respective functions are suggested to add in Introduction part.

We thank the reviewer for the helpful comment. And have added to the already existing reasons another central reason to the introduction.

Bimetallic HMF reduction electrocatalysts, in particular bimetallic oxides in the form of their oxide-derived surface-roughened catalyst analogs that evolve under reducing operando conditions, are essentially unexplored.

We demonstrate that these conditions enable the use of CuO/MO_x (M=first row transition metals) bimetallic oxide electrocatalyst precursors, which – under the reductive reaction conditions - transform into operating oxide-derived partially metallic (OD)-Cu/MO_x catalysts. In particular, the metal oxides NiO, Fe₂O₃ and Co₃O₄ were added to CuO to tune the resulting HER activity of the two-phase system, by means of tuning the surface atomic H_{ad} coverage. (Main manuscript, Page 4 and 5)

(4) For the figure 2 in page 7, since the Cu^0 and Cu^+ could be distinguished in Cu 2p XPS spectra, Cu LMM is suggested to be supplied.

We thank the reviewer for the reasonable advice and added Cu LMM to **Figure S2**.

Figure S 2: XPS measurements of CuO powder. a) Survey spectrum, b) Cu LMM Auger range c) O 1s region, d) Cu 2p region. Measurements are given in black whereas Casa XPS fits are given in light blue for O and orange for Cu.

(5) In line 164-165, the authors claim that "...a very rough Cu surface characterize by many undercoordinated Cu adatoms which likely serve as active sites for activation steps of reactant molecules...", Cu K-edge EXAFS is suggested to be added to acquire relevant local coordination information.

To support our statement and to follow the valuable suggestion, we have included citations of own and others' XANES and EXAFS studies for CuO under reductive conditions as references in the main manuscript. These works have demonstrated the sustained low averaged Cu coordination number in an oxide-derived state, evidencing the undercoordinated nature of the resulting Cu surface.

They offer rough Cu surfaces, characterized by low average geometric coordination numbers, suggesting a large fraction of undercoordinated Cu adatoms which likely serve as active sites for activation steps of reactant molecules^{12,14,15} (Main manuscript, Page 6)

These clearly show that one arrives at lower coordination numbers ($\text{CN} < 12$), suggesting enhanced roughness.

(6) The BET results are quite different over series catalysts, then how the factor about specific surface area on catalytic performance be excluded. In other word, how is the size effect of CuO excluded? Otherwise, the role of surface catalysis in MO_x is too speculative without excluding these factors.

We thank the reviewer for this important comment and agree that the size effect should be investigated. Unfortunately, it was not possible to make clear statements about the particle sizes by our TEM data

(**Figure S6**), because the particles agglomerated. For the same reason, it was also not possible to determine the particle size distribution.

Since we are aware of the impact the size effect might have, we tried to get more detailed results with HR-STEM analysis (**Figure S7**). However, also HR-STEM only supports our previous statement that the particle size can be classified between 5-100 nm. Comparing this size with the manufacturer's specification for the commercial CuO of <50 nm, however, a particle size effect seems negligible. Normally, one would assume that a smaller particle size would result in a higher surface area and thus more accessible active sites. This would then be visible by a higher activity. However, this is not the case, our synthesized CuO shows a higher BET surface area (**Figure 2k**) and a comparable BET corrected HMFRR activity (**Figure 3e**).

Figure S 7: HR-STEM images of the powder CuO and CuO/MO_x catalysts. 1a) and b) showing CuO, 2a) and b) showing CuO/NiO, 3a) and b) showing CuO/Fe₂O₃ and Figure 4a) and b) showing CuO/Co₃O₄. A scale is provided in the right bottom corner.

(7) For the figure 2 in Page 11, the error bars are lacked. In theory, the experiments should be repeated at least 3 times to be conformable with presented results.

The indication of errors should indeed be included, we thank the reviewer for this comment. We share the same opinion as the reviewer, why we repeated our experiments at least 4 times. Due to the high reproducibility, small relative errors of 2-4% for the FE_{products} and 3-5% for FE_{H₂} resulted. Therefore, and to preserve both clarity and aesthetics, we did not include the error bars in **Figure 4**. We show this as an example in **Figure S18**. Nevertheless, it is important for the reader to know how high the deviations are, so we have included them in the caption of **Figure 4**. In addition, we added **Figure S18** to the SI.

Figure S 18: MEA-Flow-Cell performance measurements of CuO/MO_x/CF electrodes (with error bars). a)-e) Faradaic efficiencies for H₂ (strong blue), BHMF (light blue), MFA (green) and other products (grey) of the different spray-coated CuO/MO_x/CF catalysts. f) Faradaic efficiencies and HMF conversion (yellow) are calculated every 15 min over 60 min using CuO/NiO/CF as a catalyst. Product color code stays as in a)-e). g)-i) Scatter plot of the product selectivity preference. HMF conversion over MFA/BHMF selectivity ratio, calculated by $S_{MFA/S_{BHMF}}$ for all CuO/MO_x catalysts on CF at different current densities for 30 min. The color code stays the same as before. Cell reaction conditions: 0.1 M KOH with 10 mM HMF as catholyte (100 mL), 0.1 M KOH as anolyte (100 ml), 5 cm² electrode area, nickel foam (NF) as anode and a flow rate of 25 mL min⁻¹, at 10-30 mA cm⁻² for 30 min. Error bars with relative errors of 2-4% for FE_{products}, 3-5% for FE_{H₂} and 1-3% for X_{HMF} were added.

(8) The mechanistic studies are very speculative and lack of support by either experiments or literatures (even if they are for thermocatalytic reaction, some information can be inferred). Particularly the C-OH group of the BAMF could be directly adsorbed into the MO_x, this is not considered in this work but likely an important path. The authors focus the competition between the $2e^-/2H^+$ -CHO hydrogenation to -CH₂OH and its subsequent $2e^-/2H^+$ hydrogenolysis to -CH₃, it is better to perform DFT calculation aiming to provide more convincing evidence. In addition, series in-situ experiments such as in situ FTIR should be designed to investigate the adsorption/activation behavior of substrate or intermediated products molecules.

We thank the reviewer for this comment. We agree that detailed Quantum chemical calculations of reaction pathways would be highly desirable in this case. Unfortunately, such theoretical computational analyses are outside the scope of our research expertise, and collaborating theoretical groups, upon request, considered this too formidable a task, given the vast configuration space of the organic molecules and the solvent molecules. We are afraid that we have to pass on computational support for our experimental data this time.

We also agree with the reviewer that in-situ FTIR would in principle be a very suitable method to clarify further details of the reaction mechanism and surface intermediates. This is why we now performed measurements for this. However, as we describe in the following, FTIR turned out to be a suboptimal characterization technique under our chosen reaction conditions:

The correct choice of the prism was very difficult, because the prism crystal had to be stable at pH=13, the penetration depth had to be high enough for adequate signal intensity and the refractive index had to be higher than that of the electrolyte, HMF and its products. The high pH value alone limits the possibilities already very much to for example diamond, germanium, silicon or CaF₂.¹⁶ Germanium, however, also falls out due to the low penetration depth. Even at different mirror angles, the signal intensity remains too weak. Even though a diamond prism has the best properties for our requirements, we could not use it because no supplier produces a diamond prism in the size and shape, we would need for our setup. In the end, only the Si (pH stability marginal) and CaF₂ prisms remained. The refractive index of CaF₂ varies between 1.39-1.42. Comparatively, this refractive index is below that of for example HMF with 1.56. Thus, actually also the Si and CaF₂ prisms are not optimal options.¹⁶ Nevertheless, we have performed measurements with these prisms (**Figure R 2**).

For this purpose, it was necessary to perform a background measurement with KOH so that the OH band of the HMF and the BHMF would not be masked by the electrolyte signal. For the CaF₂ prism it was not possible to obtain adequate measurement results, as shown in **Figure R 2a**. The Si prism showed characteristic signals that could be assigned to HMF, but only at strongly increased HMF concentration of 1M HMF (**Figure R 2b**). After 30 minutes of electrolysis at 20 mA cm⁻² with the CuO/CF catalyst, however, it should have been visible that at least the signal of the C=O band decreases in favor of the OH band, since the hydrogenation of HMF to BHMF should have been observed here. After 30 minutes reaction time and the high HMF concentration, the formation of by-products such as humins also becomes problematic.

Finally, it was not possible to perform meaningful and reproducible measurements with the FTIR.

Figure R 2: FTIR measurements with a CaF₂ and a Si prism. a) FTIR measurements with a CaF₂ prism in 0.1 M KOH with (yellow) and without (blue) 10 mM HMF with KOH background. b) FTIR measurements with a Si prism in 0.1 M KOH with 10 mM (black) and 1M (yellow to orange) HMF during electrolysis at 20 mA cm⁻² with KOH background.

(9) How about the catalytic performance in comparison with peer work especially with other non-noble catalysts. To highlight the performance advantages of OD-Cu/MOx, the evaluation results of single factors such as substrate conversion, product selectivity and other reaction condition especially temperature should be expressed through a radar graph in SI.

We thank the reviewer for the reasonable comment. We added a radar plot to the SI (**Figure S20**). However, we would like to add that a comparison here is difficult for several reasons. Firstly, there is no literature that uses a pH value above 10. Furthermore, the avoidance of noble metal catalysts in comparable pH ranges is also difficult. Due to the different initial concentrations of HMF, the potential at 10 mA cm⁻² should also be considered with caution. Last but not least, we have omitted the temperature comparison, because all measurements are made at room temperature, since an increase in

temperature can promote side reactions. Nevertheless, we hope to have met the demand of the reviewer to a sufficient extent.

Figure S 20: Comparison with the literature presented in a radar plot. Performance and reaction parameters like the HMF conversion (X_{HMF}), selectivity towards BHMF (S_{BHMF}) and MFA (S_{MFA}), the potential at 10 mA cm^{-2} ($E_{10\text{mA}/\text{cm}^2}$), pH and initial HMF concentration (C_{HMF}) are compared between CuO/Fe₂O₃/CF and data from the literature.¹⁷⁻¹⁹

(10) In order to further determine the advantages of the two-phase catalyst, it is recommended to assess the performance of the physical mixture of two single oxides for comparison. This necessary contrast is indispensable.

We thank the reviewer for this important comment and agree with the reviewer that this is an important addition. We have performed such a study for the CuO/Fe₂O₃ catalyst in the RDE configuration. It is visible that the activity of the physically mixed catalyst (**Figure S14**, yellow) is slightly worse than in the best case comparable to our presented co-precipitated catalyst (**Figure S14**, purple). We added text to the Main manuscript and the plot to the SI.

Page 8 main manuscript: *We can also add that our (co-)precipitation-calcination (air) synthesis protocol has slight catalytic advantages over the physical mixing of the individual precipitated metal oxides (Figure S14).*

Figure S 14: RDE three-electrode measurements of CuO/Fe₂O₃ compared to physical mixed CuO and Fe₂O₃. RDE measurements of the mixed metal oxide CuO/Fe₂O₃ (purple) and the physical mixed oxide CuO and Fe₂O₃ (yellow). All RDE LSV measurements were taken between 0 V_{RHE} to -0,6 V_{RHE} at a scan rate of 10 mV s⁻¹ in 0.1 M KOH with (solid line) and without (dashed line) 10 mM HMF at 2500 rpm with an electrode surface area of 0.19 cm² and a catalyst loading of 0.04 mg. All measurements are 100% manual internal resistance (IR) corrected.

(11) The authors claim that such two-phase structure is induced during the reaction process. The specific structural dynamic evolution needs to be clearer.

We thank the reviewer for this comment. We agree that an elucidation of this dynamic would be helpful, but at the same time we think that *in-situ* EXAFS/XAS measurements would be necessary. Unfortunately, we are not able to establish such measurements within the framework of this study. However, we further base our hypothesis on existing literature from us and other groups on comparable systems.^{12,14,15}

(12) How about the stability of OD-Cu/MO_x catalysts. Relevant reusability test should be added.

We thank the reviewer for this comment. We agree with the reviewer that stability tests are important. That's why we performed a stability test for the most MFA selective catalyst CuO/Fe₂O₃/CF (**Figure 5a**). Since the focus of this study is primarily on the proof of concept of alkaline HMF hydrogenolysis, we feel it is appropriate in this context to show the stability of the most relevant catalyst over 5 catalytic cycles.

(13) The authors mentioned “Alkaline conditions favor humin formation”, but did not give the selectivity of di- and polymerization products. Have the di- and polymerization products been detected and have the carbon balance been calculated in this work?

We thank the reviewer for this important and relevant comment. The problem in detecting these di- and polymerization products is their structural diversity.⁹ Thus, on the one hand, even with NMR, for example, it is difficult to detect and define exact structures. This makes the calibration of any measurement method practically impossible. Many of these molecules are also not commercially available in pure form, even if one can say with certainty exactly which structures are formed. However, we have performed mass spectroscopy measurements after 30 minutes of electrolysis with CuO/CF, which show that di- and polymers are formed (**Figure R 3**, orange area).

Figure R 3: MS measurements after 30 min of electrolysis with CuO/CF. Measurements are corresponding to the measurements of Figure 4b at 10 mA cm⁻². Method: LC-MS-APCI at Orbitrap

Reviewer 3:

The topic of this manuscript is relevant for Nature Communications. The manuscript is well written, the characterization methodology is sound, the authors provided enough detail to reproduce their work, and the results are very important for the electrochemical community. However, it contains a few experimental and analytical issues that the authors need to address before it can be accepted for publications. Therefore, I recommend reconsidering after doing minor revisions.

In summary, the authors explored the electrochemical reduction of 5-Hydroxymethylfurfural (HMF) to 5-Methylfurfuryl alcohol (MFA) under strongly alkaline reaction environments over oxide-derived Cu bimetallic electrocatalysts. The authors investigated the relationship between the surface catalysis CuO on NiO, Fe₂O₄, Co₃O₄ surfaces and the selectivity towards different reduction products such as 2,5-Bishydroxymethylfuran (BHMF) and hydrogen evolution reaction (HER). The authors provide evidence for a kinetic competition between the HER and hydrogenolysis of BHMF to MFA and showed that the product selectivity depends on the electrode composition and current density. However, each material had different (electrochemical) surface area (Figure 2K) and required different potentials to generate the same current density (Figure 3) when tested in Rotating Disk Electrodes (RDE) and batch cells; hence, the comparison should not be done as a function of current density but as a function of potential.

Additionally, all the electrodes showed similar electrochemical performance (but different product selectivity) when tested in the alkaline exchange membranes (AEM) flow cell. The authors did not measure or report the half-cell potential in the flow cell so this discrepancy in electrochemical performance between the flow and batch systems cannot be explained. The authors are encouraged to compare the performances at equal IR-compensated potentials between the batch and flow system to better understand the results.

For this reason, this manuscript cannot be accepted as it is, and the authors are encouraged to address the following points:

1. The authors show the XPS spectra of CuO, NiO, Fe₂O₃, and Co₃O₄. Do the authors observe any change in oxidation state after reaction?

We thank the reviewer for the important comment. Indeed, we have been able to detect a change in the oxidation state of the catalysts after electrolysis. (**Figure R 4**). **Figure R 4** a-e) shows a change in oxidation state for all metal oxides. However, after electrolysis, metallic (M⁰) and oxidic (M¹⁻³⁺) mixed states are obtained.^{1,7,20} This may be due to the fact that the reduction from oxidic to metallic was not completed during the reaction or that the catalysts are oxidized again after the reaction on air. Thus, it is difficult to make an exact statement by ex-situ XPS.

Figure R 4: XPS Measurements after electrolysis. a) and b) Cu 2p XPS measurements of CuO before electrolysis (orange), CuO/CF (black), CuO/NiO/CF (red), CuO/Fe₂O₃/CF (purple) and CuO/Co₃O₄/CF (blue) after electrolysis (ae). c) Ni 2p XPS results of CuO/NiO/CF (red) before and CuO/NiO/CF (light red) after electrolysis. d) Fe 2p XPS results of CuO/Fe₂O₃/CF (purple) before and CuO/Fe₂O₃/CF (light purple) after electrolysis. e) Co 2p XPS results of CuO/Co₃O₄/CF (blue) before and CuO/Co₃O₄/CF (light blue) after electrolysis

2. The performance of CuO/NiO/CF in Figure 3 g and f is very impressive, but it is not clear if the boost in performance comes from the CuO/NiO or NiO/CF. The authors are recommended to have a plot comparing the performances with and without HMF of CuO/CF, CuO/NiO, NiO/CF, and CuO/NiO/CF. Same for the Fe₂O₃/CF and Co₃O₄/CF systems.

We thank the reviewer for the very helpful and reasonable comment. We agree and added the data to the SI (**Figure S17**) as well as text to the main manuscript (**Main Manuscript, Page 9**).

At the same time, however, Figure S17 shows that CF alone does not necessarily lead to a generally increased performance. Here it becomes clear once again that the combined CuO/MO_x metal oxides with or without CF support bring an increase in activity.

Figure S 17: Activity comparison of CuO/MO_x, CuO/CF, MO_x/CF and CuO/MO_x/CF. comparing a) CuO/NiO (red), CuO/CF (black), NiO/CF (light red) and CuO/NiO/CF (dark red), b) CuO/Fe₂O₃ (purple), CuO/CF (black), Fe₂O₃/CF (light purple) and CuO/ Fe₂O₃/CF (shiny purple) and c) CuO/Co₃O₄ (blue), CuO/CF (black), Co₃O₄/CF (light blue) and CuO/Co₃O₄/CF (dark blue). Reaction conditions are the same as in Figure 3 for RDE and UTEC measurements.

3. What percentage of IR correction did the authors apply?

We thank the reviewer for the question and the resulting important additions. We have applied 100% manual IR correction. For clarification, we added this information to all captions.

4. The normalization based on BET can be misleading as not all the area is electrochemically active. The authors are recommended to obtain the real electroreactive surface area (ECSA) based on capacitance. If that is not available, the authors are recommended to just use the BET of the active phase, CuO, to evaluate the trends.

We thank the reviewer for the important comment and agree that the exact determination of the ECSA would provide even more information here. Unfortunately, due to the complexity caused by the catalysts in combination with HMF in the electrolyte, we were not able to determine it or produce reliable results. We also believe that the bifunctionality of reducing HMF and forming protons and hydrogen at the same time could lead to falsification of the ECSA. Therefore, we follow the suggestion of the reviewer and have corrected all CuO/MO_x polarization curves with the BET surface area of CuO (**Figure R 5**). However, all potential curves are corrected with the same value, which preserves the trend of **Figure 3c** and only increases the current density.

Figure R 5: RDE three-electrode measurements of CuO/MO_x. CuO BET corrected current densities of the corresponding plot from Figure 3c). All RDE LSV measurements were taken between 0 V_{RHE} to -0,6 V_{RHE} at a scan rate of 10 mV s⁻¹ in 0.1 M KOH with (solid line) and without (dashed line) 10 mM HMF at 2500 rpm with an electrode surface area of 0.19 cm² and a catalyst loading of 0.04 mg. All measurements are 100% manual internal resistance (IR) corrected.

5. The authors report the electrochemical performances (cell potentials and H₂ rates) of the MEA-flow cell for the different systems in Figure S16; that is a) CF, b) CuO/CF, c) CuO/NiO/CF, d) CuO/Fe₂O₃/CF, and e) CuO/Co₃O₄/CF. The authors show that CF at 10 mA/cm² nearly identical activity as the other materials. Similar operation voltages were observed at 20 mA/cm² for all the systems as well. At 30 mA/cm², all the systems appeared to have better performance (more stable cell voltages) than the CF itself. Hence, it is unclear what the CuO/M_xO_y is doing at 10 and 20 mA/cm², and the authors are not encouraged to draw conclusions from these experiments.

We thank the reviewer for the comprehensible comment. Since a lot of data is presented in **Figure S19** (formerly **Figure S16**), we try to explain that the performances are different, even if slightly, already at 10 and 20 mA cm⁻². Basically, an increasing HER rate is accompanied by an increasing cell potential. This is also obvious since especially CF but also CuO require high overpotentials for pure HER without HMF (**Figure 3**).

We agree with the reviewer that at 10 mA cm⁻² all catalysts are in a comparable potential range. However, the performance additionally weighted by X_{HMF}, FE_x and S_x is different (**Figure 4**). This is

not contradictory because at 10 mA cm^{-2} the HER rate is low for all catalysts and thus the larger potential driver (HER) in our opinion is comparatively insignificant and the cell potential of the different catalysts behaves comparably. Furthermore, at this current density it is not so important for the cell potential whether BHMF or MFA is preferentially formed as long as the HMF concentration is still high and HMF is primarily converted.

At 20 mA cm^{-2} , however, a potential difference becomes more obvious. Here, the cell potentials of the catalysts with higher HER activity (Fe_{H_2} , **Figure 4**) such as CuO/CF, CuO/NiO/CF and CuO/ Co_3O_4 /CF are already slightly higher than those of CF at the beginning. The cell potential and the Fe_{H_2} of CuO/ Fe_2O_3 /CF are more comparable to CF at 20 mA cm^{-2} . However, in the last minutes of the electrolysis, and hence at low HMF concentration, the cell potential for CF increases sharply and is thus significantly higher than for all other catalysts. Due to the decreasing HMF concentration, the H_2 production at the CF catalyst now also increases, whereas the other catalysts remain more selective for HMF and therefore also work at lower potentials.

We think that we included these correlations in the discussion part of the main manuscript. However, we wanted to avoid such a detailed analysis of **Figure S19** in the main manuscript in order not to digress too far. We hope to comply with the will of the reviewer.

6. The fact that the RDE and batch experiments (Fig 3) systems gave such different performances for the different electrodes evaluated while the experiments in flow cell had similar performances suggest that the flow system is not kinetically limited, and the performance might be controlled by different effects such as IR or mass transfer. The authors should measure the half-cell potential in the flow cell to better correlate the performance of the flow system with that of the RDE and batch systems.

We thank the reviewer for this important comment and agree that the determination of the half-cell potential by inserting a reference electrode would be helpful. Unfortunately, due to the MEA configuration and the specifications of the cell setup, e.g. the thickness of the gasket, it is not possible for us to place a reference electrode in the system in such a way that reliable results are generated. Therefore, we tried to bridge the gap between three electron measurements and the MEA flow cell as good as possible with the UTEC configuration. Although we are aware or agree with the reviewer that, for example, the electrolyte flow and other specifications of the MEA flow cell can influence the half-cell potential, we believe we have found a good approximation with the UTEC configuration.

7. In Figure 4, the authors are comparing the electrochemical performances at different current densities in the flow MEA cell but Figure S16 show that all the electrodes operated at nearly identical conditions (full cell voltage and H_2 rate with and without HMF). Without knowing the half-cell potential, it is difficult to rationalize whether the changes in product selectivity (Fig 4) are caused by differences in (i) potential as previously reported (e.g., ACS Catal. 2019, 9, 9964-9972 and ACS Sustainable Chem. Eng. 2020, 8, 4407-4418) or (ii) electrode composition.

We thank the reviewer for the important comment. And agree that the half-cell potential can have an impact on product selectivity. However, in our opinion, the HER is the more important factor here. Looking at the H_2 production rates without HMF, they are quite comparable. However, when HMF is added to the reaction, the H_2 rates are different. For example, CuO/ Fe_2O_3 /CF shows the highest H_2 production rate without HMF and the lowest with HMF. At the same time, CuO/ Fe_2O_3 /CF shows the lowest cell potential. These observations confirm the data from **Figure 3** that HMFRR takes place at lower potentials and also suggest that with CuO/ Fe_2O_3 /CF more adsorbed protons are selectively used for HMFRR instead of HER. Assuming that the NF on the anode always makes comparable contributions to the cell potential at the used current densities, this would therefore imply that CuO/ Fe_2O_3 /CF has a lower half-cell potential compared to the other catalysts while making HMFRR more selective. This would also be in agreement with the data from **Figure 3**. As already described in comment 6, we can't introduce a reference electrode into the MEA flow cell system, we nevertheless believe that in this case the argumentation is conclusive even without such data.

8. It is unclear how the authors can conclude that “CuO/Fe₂O₃/CF showed potential for photo-electrochemical applications with maximum currents of up to 10 mA” as the authors never explored photo-electrochemical activities in this work nor mentioned it anywhere else in the manuscript. I think this line should be removed from the conclusion section.

We thank the reviewer for this advice and removed this line from the text.

9. Figure S1 appear to contain the same information as Figure 1. I think Figure S1 should be deleted.

We thank the reviewer for this advice and removed **Figure S1** from the SI.

10. Typos:

- Line 105. “:” should be a “.”
- Line 219: should “anodic” be changed to “cathodic”?
- Line 235: should “cm-2” be “cm²”
- Line 235: Cu foam already defined in line 233
- Line 250 should say Figure 3 instead of Figure 1. It is fine in the word doc but not in the pdf doc
- Line 332 should say Figure 4 instead of 2. It is fine in the word doc but not in the pdf doc.
- Line 355 should say Figure 5 instead of 3. It is fine in the word doc but not in the pdf doc.
- Line 386 replace “Alkaline Membrane HMF Electrolyzer” for “AME”

We thank the reviewer for carefully reading and corrected the typos.

- 1 Biesinger, M. C., Lau, L. W. M., Gerson, A. R. & Smart, R. S. C. The role of the Auger
parameter in XPS studies of nickel metal, halides and oxides. *Physical Chemistry Chemical
Physics* **14**, 2434-2442, doi:10.1039/C2CP22419D (2012).
- 2 Komath, M., Thomas, S., Cherian, K. A. & Ray, A. Preparation of gamma-nickel oxyhydroxide
through high temperature reaction of nickel with anhydrous sodium hydroxide. *Materials
Chemistry and Physics* **36**, 190-193, doi:[https://doi.org/10.1016/0254-0584\(93\)90032-H](https://doi.org/10.1016/0254-0584(93)90032-H)
(1993).
- 3 Liu, L., Zhou, Z. & Peng, C. Sonochemical intercalation synthesis of nano γ -nickel
oxyhydroxide: Structure and electrochemical properties. *Electrochimica Acta* **54**, 434-441,
doi:<https://doi.org/10.1016/j.electacta.2008.07.055> (2008).
- 4 Kotta, A., Kim, E.-B., Ameen, S., Shin, H.-S. & Seo, H. K. Communication—Ultra-Small NiO
Nanoparticles Grown by Low-Temperature Process for Electrochemical Application. *Journal
of The Electrochemical Society* **167**, 167517, doi:10.1149/1945-7111/abcf51 (2020).
- 5 Li, R. *et al.* Insights into Correlation among Surface-Structure-Activity of Cobalt-Derived Pre-
Catalyst for Oxygen Evolution Reaction. *Advanced Science* **7**, 1902830,
doi:<https://doi.org/10.1002/advs.201902830> (2020).
- 6 Cong, Y., Chen, M., Xu, T., Zhang, Y. & Wang, Q. Tantalum and aluminum co-doped iron
oxide as a robust photocatalyst for water oxidation. *Applied Catalysis B: Environmental* **147**,
733-740, doi:<https://doi.org/10.1016/j.apcatb.2013.10.009> (2014).
- 7 Biesinger, M. C. *et al.* Resolving surface chemical states in XPS analysis of first row transition
metals, oxides and hydroxides: Cr, Mn, Fe, Co and Ni. *Applied Surface Science* **257**, 2717-2730,
doi:<https://doi.org/10.1016/j.apsusc.2010.10.051> (2011).
- 8 Pierre, J., Atanassov, P., Datye, A. & Goeke, R. Model Electrode Structures for Studies of
Electrocatalyst Degradation. *ECS Transactions* **33**, doi:10.1149/1.3484534 (2010).
- 9 Kloth, R., Vasilyev, D. V., Mayrhofer, K. J. J. & Katsounaros, I. Electroreductive 5-
Hydroxymethylfurfural Dimerization on Carbon Electrodes. *ChemSusChem* **14**, 5245-5253,
doi:<https://doi.org/10.1002/cssc.202101575> (2021).
- 10 Mistry, H., Varela, A. S., Kühl, S., Strasser, P. & Cuenya, B. R. Nanostructured electrocatalysts
with tunable activity and selectivity. *Nature Reviews Materials* **1**, 16009,
doi:10.1038/natrevmats.2016.9 (2016).
- 11 Tahira, A., Ibupoto, Z. H., Willander, M. & Nur, O. Advanced Co₃O₄-CuO nano-composite
based electrocatalyst for efficient hydrogen evolution reaction in alkaline media. *International
Journal of Hydrogen Energy* **44**, 26148-26157,
doi:<https://doi.org/10.1016/j.ijhydene.2019.08.120> (2019).
- 12 Wang, X. *et al.* Morphology and mechanism of highly selective Cu(II) oxide nanosheet catalysts
for carbon dioxide electroreduction. *Nature Communications* **12**, 794, doi:10.1038/s41467-021-
20961-7 (2021).
- 13 Elsayed, I., Jackson, M. A. & Hassan, E. B. Hydrogen-Free Catalytic Reduction of Biomass-
Derived 5-Hydroxymethylfurfural into 2,5-Bis(hydroxymethyl)furan Using Copper-Iron
Oxides Bimetallic Nanocatalyst. *ACS Sustainable Chemistry & Engineering* **8**, 1774-1785,
doi:10.1021/acssuschemeng.9b05575 (2020).
- 14 Timoshenko, J. & Roldan Cuenya, B. In Situ/Operando Electrocatalyst Characterization by X-
ray Absorption Spectroscopy. *Chemical Reviews* **121**, 882-961,
doi:10.1021/acs.chemrev.0c00396 (2021).
- 15 Möller, T. *et al.* Electrocatalytic CO₂ Reduction on CuOx Nanocubes: Tracking the Evolution
of Chemical State, Geometric Structure, and Catalytic Selectivity using Operando
Spectroscopy. *Angewandte Chemie International Edition* **59**, 17974-17983,
doi:<https://doi.org/10.1002/anie.202007136> (2020).
- 16 Shimadzu. *FTIR Talk Letter*,
<https://www.shimadzu.eu.com/sites/shimadzu.seg/files/firtalklettervol_11.pdf> (2009).
- 17 Sanghez de Luna, G. *et al.* AgCu Bimetallic Electrocatalysts for the Reduction of Biomass-
Derived Compounds. *ACS Applied Materials & Interfaces* **13**, 23675-23688,
doi:10.1021/acsmi.1c02896 (2021).
- 18 Zhang, Z. *et al.* Operando Generated Copper-based Catalyst Enabling Efficient Electrosynthesis
of 2,5-Bis(hydroxymethyl)furan. *Fundamental Research*,
doi:<https://doi.org/10.1016/j.fmre.2022.01.016> (2022).

- 19 Liu, H., Lee, T.-H., Chen, Y., Cochran, E. W. & Li, W. Paired electrolysis of 5-(hydroxymethyl)furfural in flow cells with a high-performance oxide-derived silver cathode. *Green Chemistry* **23**, 5056-5063, doi:10.1039/D1GC00988E (2021).
- 20 Biesinger, M. C., Lau, L. W. M., Gerson, A. R. & Smart, R. S. C. Resolving surface chemical states in XPS analysis of first row transition metals, oxides and hydroxides: Sc, Ti, V, Cu and Zn. *Applied Surface Science* **257**, 887-898, doi:<https://doi.org/10.1016/j.apsusc.2010.07.086> (2010).

REVIEWER COMMENTS

Reviewer #1 (Remarks to the Author):

The reviewer concerns on the manuscript has been well established. The manuscript can be accepted in this journal.

Reviewer #3 (Remarks to the Author):

Summary: Revisions needed

Comments:

The topic of this manuscript is relevant for Nature Communications. The manuscript is well written, the characterization methodology is sound, and the authors provided enough detail to reproduce their work. The authors addressed some of the concerns that the reviewers identified in the first review and provided additional information. However, there still appears to be some experimental flaws that the authors need to address before the manuscript can be accepted for publication. Therefore, I recommend reconsidering after doing the recommended revisions.

In summary, the authors explored the electrochemical reduction of 5-Hydroxymethylfurfural (HMF) to 5-Methylfurfuryl alcohol (MFA) under strongly alkaline reaction environments over oxide-derived Cu bimetallic electrocatalysts. The authors investigated the relationship between the surface catalysis of MO_x (NiO, Fe₂O₄, Co₃O₄) surfaces tune the selectivity of CuO towards different reduction products such as 2,5-Bishydroxymethylfuran (BHMF) and hydrogen evolution reaction (HER). The authors provide evidence for a kinetic competition between the HER and hydrogenolysis of BHMF to MFA and showed that the product selectivity depends on the electrode composition and current density. However, the trend in performance for the CuO/MO_x systems changed when testing in RDE and undivided batch systems; hence, it is unclear if the reported change in performance is associated with the secondary metal, CuO/MO_x deposition on the carbon felt (CF) for batch testing, or electrochemical reaction conditions.

The CuO/MO_x electrodes were also tested in an alkaline exchange membranes (AEM) flow cell and showed different performance to that on the RDE and undivided batch system. The CuO/MO_x systems were operated at high HMF conversions (>57% for 10 mA/cm², >85% for 20 mA/cm², and >92% for 30

mA/cm²); hence, the discussion of changes in selectivity with current is not appropriate because the main reagent is mostly consumed, and the system is “forced” to do side reactions to produce a constant rate (current). As the RDE and batch experiment show in Figure 3, the CuO/MO_x systems operated at different potentials to generate the same cell with up to 0.3 V difference. Hence, the difference in performance as a function of electrode composition shown in Figure 4 for the flow cell can also be also (partially) due to potential effects. For this reason, the authors cannot conclude that the differences are due to the nature of the secondary metal oxide.

For these reasons, this manuscript cannot be accepted as it is, and the authors are encouraged to address the following points:

Electrode synthesis and characterization

1. The authors provided the performance of physically mixed oxides in the revised submission (Figure S14), which shows that physical mixing CuO₂ and Fe₂O₃ provides similar performance as co-precipitated CuO₂/Fe₂O₃. Does this also happen for the other key CuO/NiO/CF, d) CuO/Fe₂O₃/CF, and e) CuO/Co₃O₄?
2. If the performance of the co-precipitated and physically mixed system is similar, the authors need to provide the characterization of the physically mixed system as well.

Electrochemical performance of the RDE and batch system

1. The authors provided the IR-corrected performance of the different CuO/MO_x material in the revised manuscript RDE and batch system, Figure 3c and g. The IR and mass transfer effects are completely different in both systems, so it is hard to do a clean comparison of the absolute values and the trends should be compared instead. After close examination, the trends do not always appear to be the same for the two reaction systems. For example, CuO,HMF (deposited in CF) in batch had the worst performance in batch but CuO,HMF had the second-best performance in RDE. Similarly, CuO/NiO (deposited in CF) had the best performance in batch cell (with and without HMF) while CuO/Fe₂O₃,HMF had the best performance in RDE. Are the changes in trends associated with the deposition of the metal on the CF for batch (and flow) testing or is it experimental error? The authors must address this difference in trend.
2. Additionally the shape of the curves shown in Figure 3c of some CuO/MO_x systems (CuO,HMF, CuO/NiO,HMF and CuO/Co₃O₄,HMF as well as CuO/Fe₂O₃ to some extent) in the RDE do not follow the traditional exponential trend. Hence, the authors are encouraged to revise the data and perform different IR corrections to match the data between the RDE and batch cell.

Electrochemical performance of the AEM-flow system

1. The authors indicated in the rebuttal letter that they cannot place a reference electrode in the flow cell. Per Figure 3g, there is a 0.3 V difference between the CuO/MO_x electrodes to generate the same current, so the potential at which the electrodes operated in the flow-cell to generate the same current is also different. How do the authors know that the small difference in performance observed with different secondary metal at 10 mA/cm² is caused by the nature of the secondary metal and not by operating at each system at different potentials? The authors must address this critical point. Since the half-cell potential cannot be measured, the authors are recommended to operate the flow system at different currents (i.e., potentials) but similar conversion levels to demonstrate whether the current/potential affect the product selectivity for a given CuO/MO_x combination.

2. Figures 4a to 4e and Table S3 summarizes the performance of the flow cell after 30 min of reaction. All the catalytic systems were operating at ≈90% HMF conversion at 20 mA/cm²; hence, the discussion of changes in selectivity with current is not appropriate because the main reagent is mostly consumed, and the system is “forced” to do side reactions (H₂) to produce a constant rate (current). The authors should operate the system at a similar, lower conversion level (≈10%) for all the test to truly observe the effects of current. The authors are encouraged to run the system with 10X higher concentration of HMF, lower catalyst loadings, and/or lower current densities as well.

3. Figure 4c shows the selectivity of CuO/NiO/CF at 10, 20, 30 mA/cm² after 30 min of reaction and Figure 4f shows the selectivity of the CuO/NiO/CF at 20 mA/cm² at 0-15, 15-30, 30-45, and 45-60 min. This reviewer would expect the selectivity of CuO/NiO/CF in Figure 4c at 20 mA/cm² and 30 min to be the same as the selectivity in Figure 4f at 30 min and 20 mA/cm² but it is not. Figure 4c shows a 20% H₂ and 27% MFA selectivity while Figure 4f shows 71% H₂ and 0% MFA selectivity. Why are the results so different even when the electrode, current, and reaction time are the same? The authors must address this inconsistency.

4. Figure 4 g-l show that there were small to negligible effect of the nature of the secondary metal support on CuO performance at 20-30 mA/cm². For example, at 10 mA (Figure 4g) CuO/NiO/CF and CuO/Fe₂O₃/CF are at the opposite side of the plot with different SMFA/SBHMF selectivities; however, at 20 mA/cm² CuO₂/NiO/CF and CuO/Fe₂O₃/CF both have similar SMFA/SBHMF selectivities. Finally, at 30 mA/cm² both CuO₂/NiO/CF and CuO/Fe₂O₃/CF have the same SMFA/SBHMF selectivities. However, the authors the conclusion that “the nature of the secondary metal oxide will tune the HMF reduction performance of CuO”. The authors are advised to operate the system at low conversion to mitigate effect of conversion to properly investigate the role of the nature of the secondary metal on the CuO performance.

5. Figure 4 shows the performance of selected catalytic systems in the MEA-flow cell. Figure 4f shows the conversion and FE as a function of time on stream for a constant current of 20 mA/cm² and reveals

that the conversion increases from 60% at 0-15 min to >90% conversion at 15-30 min of reaction. Did the authors run the cell under continuous recycle of the electrolyte? If so, this reviewer did not catch this from the text.

6. If not running under continuous recycle, it is unclear how the conversion increases in continuous flow reactor running in a single pass configuration. The authors must address this.

Other

- The nomenclature of the catalyst names in Figure 4 is inconsistent with the text and other Figures. For example, CuO/NiO/CF is called CuO NiO CF in Figure 4c.

Rebuttal letter – Responses to Reviewers' comments

First and foremost, we would like to thank all reviewers for their constructive comments and detailed reviews.

Reviewer 1:

The reviewer concerns on the manuscript has been well established. The manuscript can be accepted in this journal.

We thank the reviewer for his positive verdict.

Reviewer 3:

Summary: Revisions needed

Comments:

The topic of this manuscript is relevant for Nature Communications. The manuscript is well written, the characterization methodology is sound, and the authors provided enough detail to reproduce their work. The authors addressed some of the concerns that the reviewers identified in the first review and provided additional information. However, there still appears to be some experimental flaws that the authors need to address before the manuscript can be accepted for publication. Therefore, I recommend reconsidering after doing the recommended revisions.

In summary, the authors explored the electrochemical reduction of 5-Hydroxymethylfurfural (HMF) to 5-Methylfurfuryl alcohol (MFA) under strongly alkaline reaction environments over oxide-derived Cu bimetallic electrocatalysts. The authors investigated the relationship between the surface catalysis of MOx (NiO, Fe₂O₄, Co₃O₄) surfaces tune the selectivity of CuO towards different reduction products such as 2,5-Bishydroxymethylfuran (BHMF) and hydrogen evolution reaction (HER). The authors provide evidence for a kinetic competition between the HER and hydrogenolysis of BHMF to MFA and showed that the product selectivity depends on the electrode composition and current density. However, the trend in performance for the CuO/MOx systems changed when testing in RDE and undivided batch systems; hence, it is unclear if the reported change in performance is associated with the secondary metal, CuO/MOx deposition on the carbon felt (CF) for batch testing, or electrochemical reaction conditions.

The CuO/MOx electrodes were also tested in an alkaline exchange membranes (AEM) flow cell and showed different performance to that on the RDE and undivided batch system. The CuO/MOx systems were operated at high HMF conversions (>57% for 10 mA/cm², >85% for 20 mA/cm², and >92% for 30 mA/cm²); hence, the discussion of changes in selectivity with current is not appropriate because the main reagent is mostly consumed, and the system is "forced" to do side reactions to produce a constant rate (current). As the RDE and batch experiment show in Figure 3, the CuO/MOx systems operated at different potentials to generate the same cell with up to 0.3 V difference. Hence, the difference in performance as a function of electrode composition shown in Figure 4 for the flow cell can also be also (partially) due to potential effects. For this reason, the authors cannot conclude that the differences are due to the nature of the secondary metal oxide.

For these reasons, this manuscript cannot be accepted as it is, and the authors are encouraged to address the following points:

Electrode synthesis and characterization

1. The authors provided the performance of physically mixed oxides in the revised submission (Figure S14), which shows that physical mixing CuO and Fe₂O₃ provides similar performance as co-precipitated CuO₂/Fe₂O₃. Does this also happen for the other key CuO/NiO/CF, d) CuO/Fe₂O₃/CF, and e) CuO/Co₃O₄?

This is a good comment. We have now performed additional experiments to improve our manuscript. We have prepared physically mixed CuO/NiO and CuO/Co₃O₄ catalysts and tested them at the RDE scale. Our results evidence (as for CuO/Fe₂O₃) comparable activity for all physically mixed and co-precipitated catalysts. We have presented all these data in the new Supplementary Fig. 14 a-c.

Supplementary Fig. 14: RDE three-electrode measurements and powder XRD characterization of CuO/MO_x compared to physically mixed CuO and MO_x. a-c) RDE measurements of the mixed metal oxides CuO/NiO (red), CuO/Fe₂O₃ (purple), and CuO/Co₃O₄ (blue) compared to the physically mixed equivalents (green). All RDE LSV measurements were taken between 0 V_{RHE} to -0.6 V_{RHE} at a scan rate of 10 mV s⁻¹ in 0.1 M KOH with (solid line) and without (dashed line) 10 mM HMF at 2500 rpm with an electrode surface area of 0.19 cm² and a catalyst loading of 0.04 mg. All measurements are 100% manual internal resistance (IR) corrected. d-f) Powder XRD measurements of the mixed metal oxides CuO/NiO (red), CuO/Fe₂O₃ (purple), and CuO/Co₃O₄ (blue) compared to the physically mixed equivalents (green), CuO (black), and the CuO Tenorite reference (orange).

2. If the performance of the co-precipitated and physically mixed system is similar, the authors need to provide the characterization of the physically mixed system as well.

We fully agree with the reviewer that a characterization comparison between the co-precipitated and physically mixed catalysts would further strengthen our results. Therefore, we investigated the crystal structure of the physically mixed catalysts by powder XRD (Supplementary Fig. 14 d-f). The results show the same reflections for the two synthesis routes, reconfirming the formation of two phases instead of a two-metal oxide mix phase, with mainly CuO tenorite visible for all catalysts (Figure 2 and Supplementary Fig. 14 d-f).

Electrochemical performance of the RDE and batch system

1. The authors provided the IR-corrected performance of the different CuO/MO_x material in the revised manuscript RDE and batch system, Figure 3c and g. The IR and mass transfer effects are completely different in both systems, so it is hard to do a clean comparison of the absolute values and the trends should be compared instead. After close examination, the trends do not always appear to be the same for the two reaction systems. For example, CuO, HMF (deposited in CF) in batch had the worst performance in batch but CuO, HMF had the second-best performance in RDE. Similarly, CuO/NiO (deposited in CF) had the best performance in batch cell (with and without HMF) while CuO/Fe₂O₃, HMF had the best performance in RDE. Are the changes in trends associated with the deposition of the metal on the CF for batch (and flow) testing or is it experimental error? The authors must address this difference in trend.

We thank the reviewer for the comment. We double-checked our experimental data and can confirm that there was no experimental error. First, in Supplementary Fig. 17, we show both the influence of the different cells (RDE and UTEC) and the CF on the activity and mentioned this in the Main manuscript (page 9).

At the same time, however, **Supplementary Fig. 17** shows that CF alone does not necessarily lead to a generally increased performance. Here it becomes clear once again that the combined CuO/MO_x metal oxides with or without CF support bring an increase in activity.

Supplementary Fig. 17: Activity comparison of CuO/MO_x, CuO/CF, MO_x/CF, and CuO/MO_x/CF. Comparing a) CuO/NiO (red), CuO/CF (black), NiO/CF (light red), and CuO/NiO/CF (dark red), b) CuO/Fe₂O₃ (purple), CuO/CF (black), Fe₂O₃/CF (light purple) and CuO/Fe₂O₃/CF (shiny purple) and c) CuO/Co₃O₄ (blue), CuO/CF (black), Co₃O₄/CF (light blue) and CuO/Co₃O₄/CF (dark blue). Reaction conditions are the same as in Figure 3 for RDE and UTEC measurements. All measurements are 100% manual internal resistance (IR) corrected.

However, we would like to reiterate that there are differences in mass transport between the two cell configurations. In the RDE setup, we apply rotation while the electrode is static in the UTEC setup. By the rotation in the RDE, the HMF transport to the electrode and the products' removal is likely accelerated. In addition, it can be assumed that the transport of the produced hydrogen from the electrode to the solution is also accelerated, reducing the blockage of active sites at the catalyst. These reasons can therefore lead to a change in activity in addition to the CF.

In our new data, CuO/NiO with HMF (RDE) and CuO/NiO/CF with HMF (UTEC) show comparable performances up to 10 mA cm⁻² (Supplementary Fig. 17a). At higher current densities, we see that the CuO/NiO/CF with HMF (UTEC) shows higher performance compared to the CuO/NiO catalyst (RDE). But it must be noted that in the UTEC configuration, there is no difference between the curves with and without HMF. Here the HER also starts after 0.2 V_{RHE}. In the RDE configuration, the exponential drop of the curve, which we associate with an increasing HER, is not visible before 0.5 V_{RHE}. This difference in onset potentials is caused by the previously mentioned mass transport limitations in the UTEC.

In addition, the CF and, thus, the associated expansion of the three-dimensional surface area and the CF's porosity influence the activity. We also assume mass transport limitations as the reason for the decrease in performance from RDE to UTEC for the CuO/CF catalyst with HMF. However, compared to the CuO/NiO catalyst, CuO shows a bad HER activity which is why, for example, the increase in surface area by CF for the CuO/CF catalyst in UTEC does not show the same increase in activity as for the CuO/NiO/CF catalyst.

2. Additionally the shape of the curves shown in Figure 3c of some CuO/MO_x systems (CuO,HMF, CuO/NiO,HMF and CuO/Co₃O₄,HMF as well as CuO/Fe₂O₃ to some extent) in the RDE do not follow the traditional exponential trend. Hence, the authors are encouraged to revise the data and perform different IR corrections to match the data between the RDE and batch cell.

We agree with the reviewer that the shape of the curves in Figure 3c changes for the catalysts mentioned and no longer follows the "classical" exponential shape we all know from the textbooks. We would even extend this observation to parts of Figure 3a and b. It is striking that this only occurs for the curves measured with HMF-containing electrolyte. This is understandable since, after adding HMF to the electrolyte, it is no longer a "classic" one-reaction process system. HMFRR and HER take place simultaneously, which leads to a changed shape of the curves.

We would like to abstain from modified IR correction, as we think our 100% manual IR correction is accurate.

Electrochemical performance of the AEM-flow system

1. The authors indicated in the rebuttal letter that they cannot place a reference electrode in the flow cell. Per Figure 3g, there is a 0.3 V difference between the CuO/MO_x electrodes to generate the same current, so the potential at which the electrodes operated in the flow-cell to generate the same current is also different. How do the authors know that the small difference in performance observed with different secondary metal at 10 mA/cm² is caused by the nature of the secondary metal and not by operating at each system at different potentials? The authors must address this critical point. Since the half-cell potential cannot be measured, the authors are recommended to operate the flow system at different currents (i.e., potentials) but similar conversion levels to demonstrate whether the current/potential affect the product selectivity for a given CuO/MO_x combination.

We agree that different catalysts lead to different cell potentials at similar constant current densities. We have chosen a galvanostatic reaction technique and applied constant current density for 30 min reaction time to ensure that the same amount of charge is transferred to each electrode. With a potentiostatic technique, we keep the cell potential constant, but the current and, thus, the amount of charge varies. We would have a similar problem if we measured up to different conversion levels. Each catalyst would need a different reaction time to reach a given conversion at constant current density, and accordingly, a different amount of charge would be transferred.

However, in the following table (Table R1), we have summarized the performance parameters for all three current densities to show that the conversions and the cell potentials (after 30 min) are in a comparable range for the different electrodes (catalysts) at the given current densities. It should be noted that a comparison of the cell potentials is only permissible under the assumption that the other components of the cell, such as the counter electrode or membrane, always make the same contribution to the cell potential.

Table R1: Summarized performance parameters of the different electrodes in the flow cell setup.

10/20/30 mA cm⁻²	CuO/CF	CuO/NiO/CF	CuO/Fe₂O₃/CF	CuO/Co₃O₄/CF
X_{HMF} [%]	77/89/92	69/95/92	72/99/99	57/85/93
FE_{BHMF} [%]	22/22/20	26/45/27	10/39/35	6/31/21
FE_{MFA} [%]	27/36/23	14/27/23	84/44/27	51/26/10
FE_{H₂} [%]	27/29/35	43/20/43	0/17/31	9/30/32
E_{Cell} [V]	-1.89/-2.12/-2.23	-1.90/-2.16/-2.23	-1.87/-2.02/-2.11	-1.98/-2.17/-2.25

We would like to emphasize once again that the cell potentials, as well as the conversions, differ from each other within a reasonable range. Thus the statement about the change in activity due to the second metal is and remained valid in our view.

2. Figures 4a to 4e and Table S3 summarizes the performance of the flow cell after 30 min of reaction. All the catalytic systems were operating at $\approx 90\%$ HMF conversion at 20 mA/cm^2 ; hence, the discussion of changes in selectivity with current is not appropriate because the main reagent is mostly consumed, and the system is "forced" to do side reactions (H_2) to produce a constant rate (current). The authors should operate the system at a similar, lower conversion level ($\approx 10\%$) for all the test to truly observe the effects of current. The authors are encouraged to run the system with 10X higher concentration of HMF, lower catalyst loadings, and/or lower current densities as well.

We thank the reviewer for this reasonable comment and agree that with decreasing HMF concentration, the FE_{HER} increases (Figure 4f). We also agree that a constant high HMF concentration ($X_{\text{HMF}} \leq 10\%$) must be provided over the entire reaction time for more fundamental investigations. In order to measure constantly at such a conversion in our cell without changing the reaction conditions, our reaction time would be greatly reduced to less than 5 min, which at a flow of 25 ml min^{-1} could mean that the reaction would end before the electrolyte ($V=100 \text{ ml}$) has completely flowed through the cell once. However, a 10x higher HMF concentration is not an option since at high HMF concentrations in alkaline electrolytes, parasitic side reactions to humins occur more frequently.^{1,2} Thus, our initial HMF concentration would steadily decrease and quickly fall below the 10% conversion limit.

Based on our loading study (Supplementary Fig. 16), lowering the catalyst loading at 10 mA cm^{-2} isn't an option either because it would lead to increased cathodic half-cell potentials and, thus, to a more dominant HER.

Lowering the current density to, for example, 5 mA cm^{-2} would indeed counteract a too-rapid HMF conversion, a half-cell potential increase, and stronger HER. Still, measuring at such low current densities is rather unusual in a 5 cm^2 electrode flow cell.

In summary, we understand the importance of the reviewer's concerns but find them currently very difficult to impossible to implement in our study for the reasons mentioned above. Especially we also think that the flow cell configuration is not the most suitable here. At the same time, we keep these important points in mind for another study with a different focus.

3. Figure 4c shows the selectivity of CuO/NiO/CF at 10, 20, 30 mA/cm^2 after 30 min of reaction and Figure 4f shows the selectivity of the CuO/NiO/CF at 20 mA/cm^2 at 0-15, 15-30, 30-45, and 45-60 min. This reviewer would expect the selectivity of CuO/NiO/CF in Figure 4c at 20 mA/cm^2 and 30 min to be the same as the selectivity in Figure 4f at 30 min and 20 mA/cm^2 but it is not. Figure 4c shows a 20% H_2 and 27% MFA selectivity while Figure 4f shows 71% H_2 and 0% MFA selectivity. Why are the results so different even when the electrode, current, and reaction time are the same? The authors must address this inconsistency.

We thank the reviewer for the comment. The difference between these two plots is that in Figure 4c, the entire period of 30 minutes is considered, and in Figure 4f, 15-minute time intervals are considered. For a better understanding, we added more information to the caption of Figure 4f (main manuscript, page 12).

f) Faradaic efficiencies and HMF conversion (yellow) are calculated for every 15 min time interval over 60 min using CuO/NiO/CF as a catalyst.

4. Figure 4 g-I show that there were small to negligible effect of the nature of the secondary metal support on CuO performance at 20-30 mA/cm². For example, at 10 mA (Figure 4g) CuO/NiO/CF and CuO/Fe₂O₃/CF are at the opposite side of the plot with different S_{MFA}/S_{BHMF} selectivities; however, at 20 mA/cm² CuO/NiO/CF and CuO/Fe₂O₃/CF both have similar S_{MFA}/S_{BHMF} selectivities.

Finally, at 30 mA/cm² both CuO/NiO/CF and CuO/Fe₂O₃/CF have the same S_{MFA}/S_{BHMF} selectivities. However, the authors the conclusion that "the nature of the secondary metal oxide will tune the HMF reduction performance of CuO". The authors are advised to operate the system at low conversion to mitigate effect of conversion to properly investigate the role of the nature of the secondary metal on the CuO performance.

We thank the reviewer for this important comment and confirm the observations that with increasing current density, hydrogenation predominates over hydrogenolysis. We have also observed this and described it in the main manuscript (page 11).

To display the impact of the secondary oxide phase on the competition between hydrogenolysis and hydrogenation, in other words, on the MFA/BHMF selectivity preference, we plotted the HMF conversion vs the absolute S_{MFA}/S_{BHMF} ratio in Figure 4g-i. As HMF conversions rise to completion ($X_{HMF} = 1$ for CuO/Fe₂O₃/CF at 20 and 30 mA cm⁻²), the S_{MFA}/S_{BHMF} ratio follows the FE trend. Hydrogenation prevails over hydrogenolysis at higher currents and conversions. Only at 10 mA cm⁻² for CuO/Fe₂O₃/CF and CuO/Co₃O₄/CF, where no or hardly any hydrogen is formed, hydrogenolysis is preferred over hydrogenation.

After we have discussed and excluded in comment 2 possibilities like lower current densities, strongly shortened reaction time, deviating catalyst loadings, or higher HMF concentrations to keep the conversion level low, we would like to add a detection limit problem. The use of our HPLC does not allow us to determine online and time resolved HMF conversion and product yields. This would require a previous try and error study to determine an approximate reaction time for each catalyst. We expect the error to be too high, as we cannot completely prevent HMF from undergoing side reactions while waiting in the queue of the HPLC and thus falsifying the conversion results. For these reasons, we do not believe that our system can perform accurate and reproducible measurements at low HMF conversion levels that investigate the role of the secondary metal adequately.

In conclusion, however, we think that the whole of Figure 4 and the corresponding discussion proves that the second metal oxide/metal affects the HMFRR and HER activity at identical cell configuration and identically transferred charge. Even at 20 mA cm⁻², different catalysts show different performances.

5. Figure 4 shows the performance of selected catalytic systems in the MEA-flow cell. Figure 4f shows the conversion and FE as a function of time on stream for a constant current of 20 mA/cm² and reveals that the conversion increases from 60% at 0-15 min to >90% conversion at 15-30 min of reaction. Did the authors run the cell under continuous recycle of the electrolyte? If so, this reviewer did not catch this from the text.

We thank the reviewer for the important comment. For all measurements in Figure 4, the catholyte and anolyte were continuously recycled. We have added this to the caption of Figure 4 (main manuscript, page 12).

Cell reaction conditions: 0.1 M KOH with 10 mM HMF as catholyte (100 ml, recycled), 0.1 M KOH as anolyte (100 ml, recycled), 5 cm² electrode area, nickel foam (NF) as anode and a flow rate of 25 ml min⁻¹, at 10-30 mA cm⁻² for 30 min. Relative errors of 2-4% for $FE_{products}$, 3-5% for FE_{H_2} , and 1-3% for X_{HMF} resulted (Supplementary Fig. 18).

6. If not running under continuous recycle, it is unclear how the conversion increases in continuous flow reactor running in a single pass configuration. The authors must address this.

The electrolytes were continuously recycled, as described in comment 5.

Other

• The nomenclature of the catalyst names in Figure 4 is inconsistent with the text and other Figures. For example, CuO/NiO/CF is called CuO NiO CF in Figure 4c.

We thank the reviewer for the intensive review. We scanned and corrected the manuscript for inconsistencies and errors.

- 1 Liu, S. *et al.* Advances in understanding the humins: formation , prevention and application. *Applications in Energy and Combustion Science* **10**, 100062, doi:<https://doi.org/10.1016/j.jaecs.2022.100062> (2022).
- 2 Motagamwala, A. H. *et al.* Toward biomass-derived renewable plastics: Production of 2,5-furandicarboxylic acid from fructose. *Science Advances* **4**, eaap9722, doi:10.1126/sciadv.aap9722 (2018).

REVIEWERS' COMMENTS

Reviewer #3 (Remarks to the Author):

The authors have successfully addressed the concerns of this reviewer. The manuscript can be accepted as is in Nature Communications. Great job!